# Variant PCGF1-PRC1 links PRC2 recruitment with differentiation-associated transcriptional inactivation at target genes

Hiroki Sugishita [1,2,3], Takashi Kondo[1], Shinsuke Ito[1], Manabu Nakayama[4], Nayuta Yakushiji-Kaminatsui [1], Eiryo Kawakami [5,6], Yoko Koseki[1,2], Yasuhide Ohinata[1,2], Jafar Sharif[1], Mio Harachi[1], Neil P. Blackledge[7], Robert J. Klose [7] & Haruhiko Koseki [1,2 ✉]

Polycomb repressive complexes-1 and -2 (PRC1 and 2) silence developmental genes in a spatiotemporal manner during embryogenesis. How Polycomb group (PcG) proteins orchestrate down-regulation of target genes upon differentiation, however, remains elusive. Here, by differentiating embryonic stem cells into embryoid bodies, we reveal a crucial role for the PCGF1-containing variant PRC1 complex (PCGF1-PRC1) to mediate differentiation-associated down-regulation of a group of genes. Upon differentiation cues, transcription is down-regulated at these genes, in association with PCGF1-PRC1-mediated deposition of histone H2AK119 mono-ubiquitination (H2AK119ub1) and PRC2 recruitment. In the absence of PCGF1-PRC1, both H2AK119ub1 deposition and PRC2 recruitment are disrupted, leading to aberrant expression of target genes. PCGF1-PRC1 is, therefore, required for initiation and consolidation of PcG-mediated gene repression during differentiation.

[1] Laboratory for Developmental Genetics, RIKEN Center for Integrative Medical Sciences, Yokohama, Japan. [2] Cellular and Molecular Medicine, Advanced Research Departments, Graduate School of Medicine, Chiba University, Chiba, Japan. [3] International Research Center for Neurointelligence (IRCN), Institutes for Advanced Study, The University of Tokyo, Bunkyo-ku, Japan. [4] Laboratory of Medical Omics Research, Kazusa DNA Research Institute, Kisarazu, Japan. [5] Artificial Intelligence Medicine, Graduate School of Medicine, Chiba University, Chiba, Japan. [6] Healthcare and Medical Data Driven AI based Predictive Reasoning Development Unit, RIKEN Medical Sciences Innovation Hub Program, Yokohama, Japan. [7] Department of Biochemistry, University of Oxford, Oxford, UK. ✉email: haruhiko.koseki@riken.jp

Polymb group (PcG) factors are conserved among metazoans and are required for the downregulation of developmental genes. In mammals, developmental genes frequently possess CpG islands (CGIs). PcG complexes bind these CGIs and mediate down-regulation of target genes[1–4]. At the molecular level, PcG proteins form at least two distinct multi-meric protein complexes, Polycomb repressive complexes-1 (PRC1) and -2 (PRC2). PRC1 mediates mono-ubiquitination of Histone H2A at lysine 119 (H2AK119ub1) via RING1A/B, while PRC2 mediates Histone H3 lysine 27 (H3K27me) methylation via EZH1/2[5–7]. PRC1 binds to H3K27me3 through chromobox (CBX) proteins[8,9]. However, PRC1 complexes harboring RYBP or YAF2, without CBX components, occupy genomic regions independently of PRC2[10,11]. To explain this key feature, one must consider the variety of Polycomb complexes that exists in mammalian cells. Indeed, previous studies have revealed six PRC1 subcomplexes which share a common RING1A/B catalytic subunit but mutually distinct PCGF (PCGF1-6) accessory proteins[10,11]. PRC2 also forms two distinct complexes, namely, PRC2.1 and PRC2.2[12]. Such structural variety indicates a division of labor among PRC1 and PRC2 complexes, though they share a common function to mediate H2AK119ub1 or H3K27me3, respectively. This complexity likely confers robustness and reversibility to PcG-mediated gene silencing[10,12,13].

The variant PCGF1-PRC1 complex is composed of PCGF1, RING1A/B, KDM2B, BCOR, SKP1, and RYBP, and recognizes CGIs via the CXXC domain of KDM2B to mediate H2AK119ub1[1,2,14]. Local H2AK119ub1 deposition by PCGF1-PRC1 facilitates targeting of PRC2.2, which acts in synergy with PRC2.1 to regulate subsequent H3K27me3 deposition and recruitment of canonical PRC1 containing PCGF2/4 (also known as MEL18/BMI1)[15,16]. H3K27me3-dependent recruitment of canonical PCGF2/4-PRC1 mediates robust down-regulation of PRC1 target genes[17]. Given that the PCGF1-PRC1 complex can drive de novo formation of Polycomb chromatin domains containing PRC1, H2AK119ub1, PRC2, H3K27me3, and canonical PCGF2/4-PRC1, it is also plausible that the same complex plays a dynamic role to initiate PcG-mediated silencing[14]. However, whether and how PcG factors, not only PCGF1-PRC1 but also other PRC1 sub-complexes, regulate dynamic changes of target gene expression during differentiation remains largely unknown.

Here, we, therefore, designed this study to address the role of PRC1 during differentiation-associated transcriptional changes, particularly during transcriptional inactivation, of PcG target genes. By differentiating embryonic stem cells (ESCs) into embryoid bodies (EBs), we reveal a role for PCGF1-PRC1 to mediate differentiation-associated down-regulation of a group of genes. Differentiation induces down-regulation at these genes, in association with PCGF1-PRC1 mediated deposition of H2AK119ub1 and PRC2 recruitment. By depleting PCGF1, both H2AK119ub1 deposition and PRC2 recruitment are disrupted, leading to aberrant expression of the target gene. PCGF1-PRC1 is, therefore, required for initiation and consolidation of PcG-mediated gene repression during differentiation.

## Results

**A group of PRC1 target genes is down-regulated upon ESC differentiation.** Mouse ESCs undergo self-renewing proliferation while maintaining their developmental potential, and are a good model to study the diverse function of PRC1 sub-complexes. Indeed, we have previously shown that disruption of RING1A/B in ESCs leads to ectopic or premature activation of developmental genes and exit from pluripotent status[18]. To explore the role of PRC1 subcomplexes during differentiation-associated down-regulation of gene expression, we derived EBs by differentiating ESCs.

To identify the genes that are down-regulated during ESC-to-EB differentiation in a PRC1-dependent manner, we established Pcgf1 conditional ESCs from Pcgf1[fl/fl]Rosa26[CreERT2tg/+] blastocysts[19,20]. Undifferentiated ESCs were cultured in LIF (leukemia inhibitory factor) and 3i (SU5402 for FGFR, PD184352 for MEK1/2, and CHIR99021 for GSK3β), and were maintained on mouse embryonic fibroblasts (MEFs). Induction of EBs was done by removing 3i and LIF, and by culturing in suspension for 48 h (Fig. 1a). Chromatin immunoprecipitation followed by deep sequencing (ChIP-seq) revealed that during the transition from ESCs to EBs, binding of RING1B was increased (Group 1) in 164 genes, decreased (Group 3) in 73 genes, and remained unchanged (Group 2) in 1542 genes (Fig. 1b; Supplementary Data 1). RING1B binding at transcriptional start sites (TSS) of Group 1 or 3 genes are shown by meta-plots (Fig. 1c), and by visual inspection for selected Group 1 genes (i.e., Klf4, Tbx3, and Pdgfa) or Group 3 genes (i.e., Sox4 and Grhl2) (Fig. 1d). We observed an increase of RING1B binding at the CGIs of Klf4, Tbx3, and Pdgfa, but a decrease at Sox4 and Grhl2. Of note, gene ontology analysis revealed that development- or differentiation-related genes were similarly enriched among Group 1, 2, and 3 (Supplementary Fig. 1).

We next examined whether the increase or decrease of RING1B binding led to transcriptional changes. Upon ESC-to-EB differentiation, we found 765 upregulated (log2FC > 2) and 625 downregulated (log2FC < −2) genes in total (Supplementary Data 2). Consistent with the repressive role of PcG factors, Group 1 genes showed down-regulation, and Group 3 genes showed up-regulation, while Group 2 genes showed no change in expression (Fig. 1e). The increase of RING1B binding at Group 1 genes is therefore linked to transcriptional downregulation during ESC-to-EB differentiation.

**PCGF1 mediates differentiation-associated downregulation of Group 1 genes and RING1B recruitment.** To determine whether RING1 proteins (RING1A and 1B) are involved in differentiation-associated downregulation of Group 1 genes, we ablated both genes using previously published Ring1A/B double knockout (dKO) ESCs[18]. Since Ring1A/B-dKO ESCs cannot be maintained in long-term culture, we induced EB formation and RING1B depletion at the same time point (Fig. 2a, Supplementary Fig. 2b). In the absence of RING1A and 1B, Group 1 genes were not downregulated in EBs, showing the role of PRC1 to down-regulate Group 1 genes upon differentiation.

We tested if the accumulation of PRC1 (e.g., RING1B) at Group 1 genes could be mediated by specific PRC1 sub-complexes. To this end, we examined the expression of Pcgf1-6 genes during ESC-to-EB differentiation and found that all were expressed in ESC and EB (Supplementary Fig. 2a). We, therefore, systematically deleted each Pcgf gene, to identify which one (s) was responsible for the downregulation of Group 1 genes (Supplementary Fig. 2b)[19–21]. Pcgf1-5 genes were deleted two days prior to EB formation; while Pcgf6 was deleted at the same time as EB formation, given the requirement of PCGF6 for ESC maintenance. Ablation of Pcgf1 or Pcgf6 led to modest disruption in the downregulation of Group 1 genes. In contrast, there was no effect on Group 1 gene downregulation in Pcgf2 and 4 (canonical PRC1) depleted, or Pcgf3 and 5 (variant PRC1) depleted ESCs (Fig. 2a, Supplementary Fig. 2c). Intriguingly, while Pcgf1-KO exclusively affected downregulation of Group 1 genes, Pcgf6-KO caused defects in downregulation of a broad range of PcG-target genes (Groups 1 and 2).

To determine whether PCGF1 or PCGF6, was responsible for PRC1 recruitment to target genes, we investigated RING1B binding at Group 1 gene promoters during the transition from

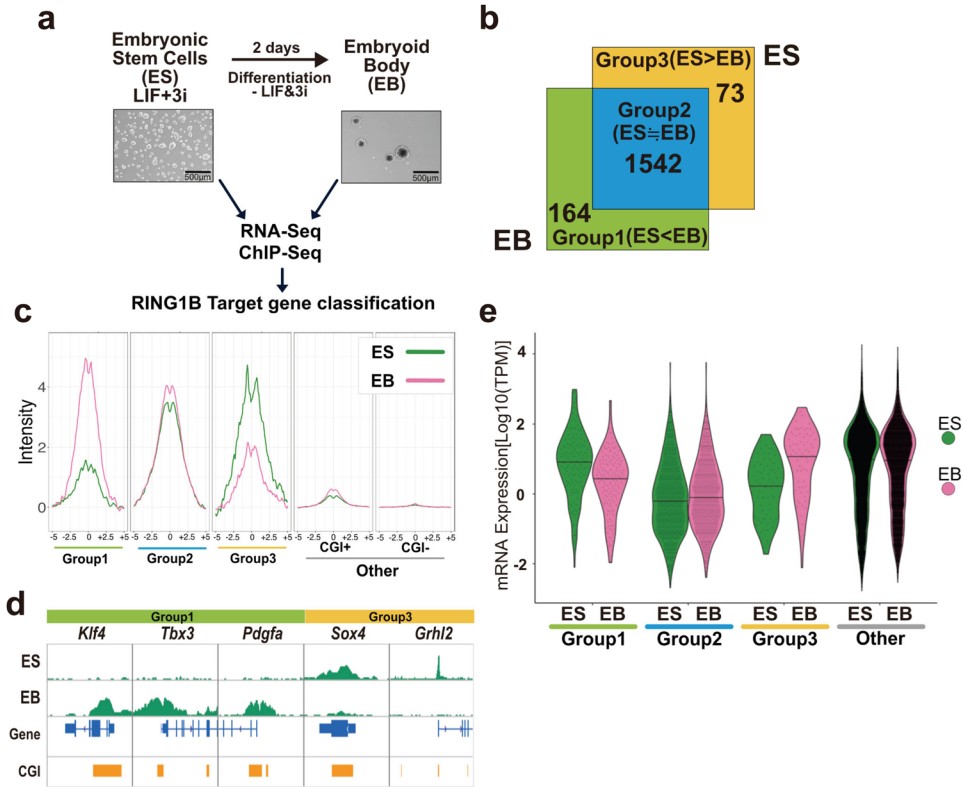

**Fig. 1 Differentiation-associated accumulation of RING1B binding at Group 1 genes takes place in association with downregulation of gene expression.**
**a** Overview of the experimental design to identify genes that are downregulated during ESC-to-EB differentiation and bound by RING1B. Representative colony morphologies for ESCs and EBs are also shown. Experiments were repeated independently at least 2 times with the same results. **b** Identification of genes at which RING1B binding changes during ESC-to-EB transition. The number of genes in each category is shown in the Venn diagram. Genes bound by RING1B are classified into "Group1" (Log2FC[EB/ES] > 1)., "Group2" ($-1 \leq$ Log2FC[EB/ES] $\leq 1$) or "Group3" (Log2FC[EB/ES] < $-1$), and are indicated by green, blue and yellow, respectively. **c** Average binding of RING1B in each group in ESCs and EBs. Metaplot views of RING1B binding at TSS ± 5 kb at each gene group are shown. Genes not bound by RING1B in both ESCs and EBs are further divided into CGI + (associated with CGIs) or CGI- (not associated with CGIs). Data from ESCs or EBs are indicated by green or pink lines, respectively. **d** Binding of RING1B at selected Group1 genes (*Klf4*, *Tbx3*, *Pdgfa*), and Group3 genes (*Sox4*, *Grhl2*), in ESCs and EBs. Snapshots of RING1B distributions in the ESCs and EBs are shown. Gene structures and positions of CpG islands (CGI) are indicated at the bottom. **e** Changes in the expression of genes differentially bound by RING1B during ESC-to-EB differentiation. Violin plot shows of the expression of genes in each group, in ESCs and EBs.

ESC to EB in *Pcgf1*-KO or *Pcgf6*-KO. In *Pcgf1*-KO, RING1B binding at Group 1 genes, but not Group 2 or 3 genes, was disrupted (Fig. 2b). In contrast, *Pcgf6*-KO did not affect RING1B binding (Supplementary Fig. 2d). Visual inspection of several Group 1 genes, such as *Klf4*, *Tbx3*, and *Pdgfa* further confirmed this result. Consistent with this observation, differentiation induced RING1B accumulation at these genes in EBs was disrupted in *Pcgf1*-KO, but unaffected in *Pcgf6*-KO (Fig. 2c, Supplementary Fig. 2e). Downregulation of RING1B binding at Group 3 genes, such as *Sox4* and *Grhl2*, did not change in either the *Pcgf1*- or *Pcgf6*-KO. To validate the role of PCGF1 to recruit RING1B, we analyzed the top-25 genes in Group 1 or 3 showing the largest change in RING1B binding during ESC-to-EB transition (Fig. 2d). The increase of RING1B binding at the top-25 Group 1 genes was PCGF1-dependent, while a decrease of RING1B binding at the top-25 Group 3 genes was not dependent on PCGF1, as expected.

To further probe the role of PCGF1 to recruit RING1B at target genes upon differentiation, we took advantage of a separate experimental model, namely, ESC to epiblast-like cells (epiLCs) differentiation (Supplementary Fig. 2f)[22]. In WT epiLCs, RING1B binding at *Klf4* and *Tbx3* was increased, compared to ESCs. However, in *Pcgf1*-KO epiLCs, RING1B binding was disrupted, showing a role for PCGF1 to recruit RING1B during transcriptional downregulation (Supplementary Fig. 2g, h).

**PCGF1 regulated the transition of transcriptional status during differentiation**. The above results indicate that PCGF1 is crucial for progressive downregulation, rather than maintenance of gene repression, at Group 1 genes. To explore this phenomenon in detail, we examined the dynamics of gene expression changes at *Klf4*, *Tbx3*, and *Pdgfa*, *Sox4*, and *Grhl2* during ESC-to-EB differentiation. We noted no obvious differences in expression levels of *Klf4*, *Tbx3*, and *Pdgfa* between WT and *Pcgf1*-KO during the first 8 h of ESC-to-EB differentiation (Supplementary Fig. 2i). Intriguingly, after the 8-hour stage, downregulation of *Klf4*, *Tbx3*, and *Pdgfa* was disrupted in the *Pcgf1*-KO. In contrast, *Sox4* and *Grhl2* genes were upregulated in both *Pcgf1*-KO and WT. The downregulation of *Klf4*, *Tbx3*, and *Pdgfa* after the 8-h stage was also accompanied by RING1B accumulation (Supplementary Fig. 2j).

To confirm that PCGF1 does not regulate maintenance of gene repression, we used the epiblast-like stem cell (epiSCs) experimental system. In both ESCs and epiSCs, key PcG targets such as *Klf4* and *Tbx3* remain repressed. Furthermore, the repressed status of these genes was unaffected in *Pcgf1*-KO epiSCs (Supplementary Fig. 2l). *Pcgf1*-KO epiSCs also did show any obvious defects in cell proliferation or colony morphology (Supplementary Fig. 2k). Taken together, our findings show that PCGF1 is required for downregulation, but not maintenance, of stable repression, of PcG target genes.

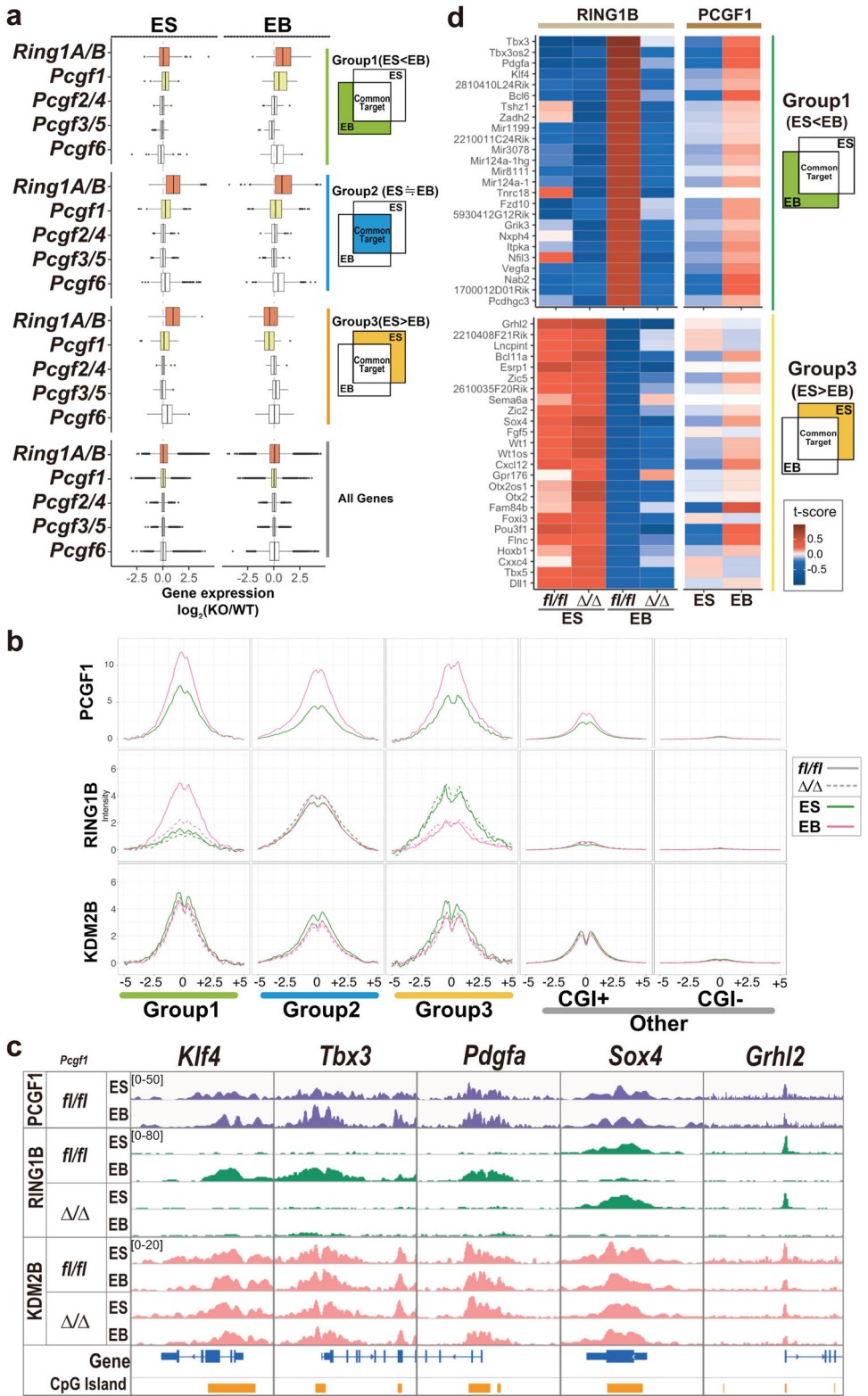

**Differentiation-associated downregulation of the Group 1 genes and RING1B recruitment is dependent on PCGF1.** We wondered how PCGF1 mediated the recruitment of PRC1 to target genes during differentiation. A previous finding showed that PCGF1 dimerizes with RING1B to form a variant PCGF1-PRC1 complex[1,10]. In addition, we reported that PCGF1-PRC1 recruits a canonical PCGF2/4-PRC1 complex to mediate H2AK119ub1 and subsequent H3K27me3 deposition[14].

Therefore, PCGF1 could directly interact with RING1B or could facilitate RING1B activity (i.e., H2AK119ub1), to downregulate Group 1 genes.

To test the above hypotheses, we asked if PCGF1 binding at the Group 1 genes was altered at the same level as RING1B during ESC-to-EB differentiation. Here, we used the anti-PCGF1 antibody that we used in our previous studies (Supplementary Table 2)[16,20]. PCGF1 bound to Group 1 genes in ESC, and this

**Fig. 2 PCGF1 is required for differentiation-associated downregulation of PRC1 target genes. a** Gene expression changes in each group (Group1: n = 164, Group2: n = 1542, Group3: n = 73) upon deletion of *Ring1a/b*, *Pcgf1*, *Pcgf2/4*, *Pcgf3/5* or *Pcgf6* in ESCs and EB. Box plot comparisons show fold changes (KO vs WT) in mRNA expression in ESCs and EBs. The interquartile range (IQR) is defined as follows: the first quartile (Q1) is greater than 25% of the data and less than the other 75%. The second quartile (Q2) sits in the middle, dividing the data in half. Q2 is also known as the median. The third quartile (Q3) is larger than 75% of the data and smaller than the remaining 25%. In a box and whiskers plot, the ends of the box and its centerline mark the locations of these three quartiles. Each whisker extends to the furthest data point in each wing that is within 1.5 times the IQR. The outlier value is > 1.5 times and < 3 times the IQR beyond either end of the box. **b** Average binding of PCGF1, RING1B, and KDM2B in each gene group in WT or *Pcgf1*-KO ESCs and EBs. Metaplot views of PCGF1, RING1B, and KDM2B binding at the TSS (5 kb upstream to 5 kb downstream) for each gene group in WT (*fl/fl*; solid lines) or *Pcgf1*-KO (Δ/Δ; dotted lines) ESCs and EBs. Data from ESCs and EBs are indicated by green and pink lines, respectively. **c** Binding of PCGF1, RING1B, and KDM2B at selected Group 1 genes (*Klf4*, *Tbx3*, *Pdgfa*) and Group 3 genes (*Sox4*, *Grhl2*) in WT or *Pcgf1*-KO ESCs and EBs. Genomic snapshots of PCGF1, RING1B, and KDM2B distributions in WT (*fl/fl*) or *Pcgf1*-KO (Δ/Δ) ESCs and EBs are also shown. Gene structures and positions of CpG islands are indicated at the bottom. **d** Binding of RING1B and PCGF1 at 25 Group 1 or 3 genes in WT or *Pcgf1*-KO ESCs and EBs. Heat map views show RING1B binding in WT (*fl/fl*) or *Pcgf1*-KO (Δ/Δ) ESCs and EBs.

binding was modestly increased upon differentiation to EB (Figs. 2b, c and 1c). RING1B binding to Group 1 genes, however, showed a stronger increase during ESC-to-EB differentiation. We also examined the binding of KDM2B, which functions as a scaffold for CGI recognition by the variant PRC1 complex. KDM2B is bound to Group 1 genes in both ESCs and EBs, but this binding was not dependent on PCGF1 (Fig. 2b, c). We noted a slight increase of PCGF1 binding at Group 2 and 3 genes as well. This binding, however, did not lead to increased RING1B recruitment. Therefore, although the modest increase of PCGF1 binding at Group 1 genes might play some role for RING1B recruitment via physical interaction between PCGF1 and RING1B, we speculated that there could be additional mechanisms for RING1B recruitment to Group 1 genes.

RING1B is also incorporated in the canonical PRC1 (cPRC1) complex, and as it is known that PRC2 binding to target genes leads to downstream recruitment of cPRC1, we wondered whether cPRC1 played a role for robust RING1B recruitment to Group 1 genes. To this end, we asked if PCGF1-PRC1 facilitated PRC2 binding, and subsequent cPRC1 binding during ESC-to-EB transition. To this end, we asked if the binding of the PRC2-component SUZ12, and the cPRC1 component PCGF2, increased upon differentiation. Indeed, we found that SUZ12 and PCGF2 progressively accumulated at *Klf4*, *Tbx3*, and *Pdgfa* after the 8-h stage of ESC-to-EB differentiation, in a similar fashion to RING1B (Supplementary Fig. 2j). Induced recruitment of RING1B at Group 1 genes during the transition from ESC to EB, therefore, could be mediated by canonical PRC1, as observed by PCGF2 binding.

**PCGF1-dependent H2AK119ub1 facilitates downregulation of the Group 1 genes and PRC2 recruitment upon differentiation.** Given that deposition of H2AK119ub1 is a major function of PRC1, we asked if H2AK119ub1 played a role in PCGF1-dependent downregulation of Group 1 genes. Curiously, in both ESCs and EBs, we detected high levels of H2AK119ub1 levels at Group 1 genes (Fig. 3a, Supplementary Fig. 3a). The same trend was seen in Group 2 and Group 3 genes. This indicated that H2AK119ub1 is deposited irrespective of the transcriptional status of the target genes. Ablation of *Pcgf1* led to decreases H2AK119ub1 deposition during ESC-to-EB transition at Group 1 genes, including *Klf4*, *Tbx3*, and *Pdgfa* (Fig. 3a, Supplementary Fig. 3a). However, a similar reduction was also seen at Group 2 and 3 genes in both ESCs and EBs (Fig. 3a, Supplementary Fig. 3a). These results indicate that PCGF1 mediates deposition of H2AK119ub1 in a broad range of PcG-targets, including Group 1 genes, consistent with a recent report[20].

In contrast, the increase of PRC2 activity (i.e., H3K27me3 deposition) at Group 1 genes such as *Klf4*, *Tbx3* during ESC-to-EB differentiation was dependent on PCGF1, indicating that

PCGF1 mediates differentiation-induced deposition of H3K27me3 (Fig. 3a). Consistent with this notion, ChIP-qPCR revealed that SUZ12, PCGF2, and RING1B binding at *Klf4*, *Tbx3*, and *Pdgfa* genes was also dependent on PCGF1 (Fig. 3b). H3K27me3 deposition, and binding of SUZ12 (PRC2), PCGF2, and RING1B (cPRC1) at *Sox4* and *Grhl2*, however, was reduced upon differentiation in a PCGF1-independent manner. Therefore, PCGF1 mediates PRC2 and subsequently PRC1 recruitment to newly establish PcG-repressive domains, as observed at Group 1 genes.

Having shown that PCGF1 is required for PRC2 and cPRC1 recruitment and function at Group 1 genes, we went on to examine whether H2AK119ub1 was involved in the downregulation of Group 1 genes and PRC2 recruitment in EBs. To this end, we took advantage of an ESCs line harboring *Ring1A* point mutant (*Ring1A*-pm) and *Ring1B* conditional point mutant (*Ring1B*-cpm) alleles, in which vPRC1-dependent H2AK119ub1 can be abrogated by 4-hydroxytamoxifen (4-OHT) treatment[16] (Fig. 3d). RNA-seq analysis revealed that Group 2 and 3 genes were de-repressed in 4-OHT treated ESCs (double catalytic dead), similar to *Ring1A/B*-dKO ESCs (Fig. 2a, Fig. 3c, Supplementary Fig. 3b)[16]. Furthermore, downregulation of Group 1 genes was perturbed in *Ring1A*-pm and *Ring1B*-cpm EBs (Fig. 3c, Supplementary Fig. 3b). Consistent with this observation, induction of H3K27me3 deposition at *Klf4*, *Tbx3*, and *Pdgfa* genes were disrupted in *Ring1A*-pm and *Ring1B*-cpm EBs (Fig. 3d). H2AK119ub1, therefore, mediates transcriptional downregulation and PRC2 recruitment at Group 1 genes.

Intriguingly, RING1B binding, although reduced compared to WT, persisted at *Klf4*, *Tbx3*, and *Pdgfa* genes in 4-OHT treated EBs, unlike H3K27me3 (Fig. 3d). This may represent the binding of RING1B incorporated in the vPRC1 (PCGF1-PRC1) complex, rather than cPRC1. We tested this possibility by disrupting H3K27me3-dependent recruitment of cPRC1, taking advantage of an EED conditional ESC line in which exon 3 of the *Eed* gene encoding the first WD40 motif could be ablated by the addition of 4-OHT. In *Eed*-cKO cells we observed global depletion of H3K27me3, as expected (Supplementary Fig. 3c, d, e, f)[23]. PCGF1, however, was recruited to the *Klf4*, *Tbx3*, and *Pdgfa* genes in *Eed*-KO ESCs and EBs, to a similar extent to WT (Supplementary Fig. 3g), revealing that PCGF1-PRC1 functions upstream of PRC2 and cPRC1 during the transition from ESCs to EBs. These results show that H2AK119ub1 deposited by PCGF1-PRC1 is required for downregulation of the Group 1 genes via PRC2-dependent H3K27me3 deposition.

**Downregulation of transcriptional activity triggers PCGF1-dependent PRC2 recruitment to CGIs.** To determine how differentiation cues activate PCGF1-dependent downregulation of gene expression during ESC-to-EB transition, we focused on the

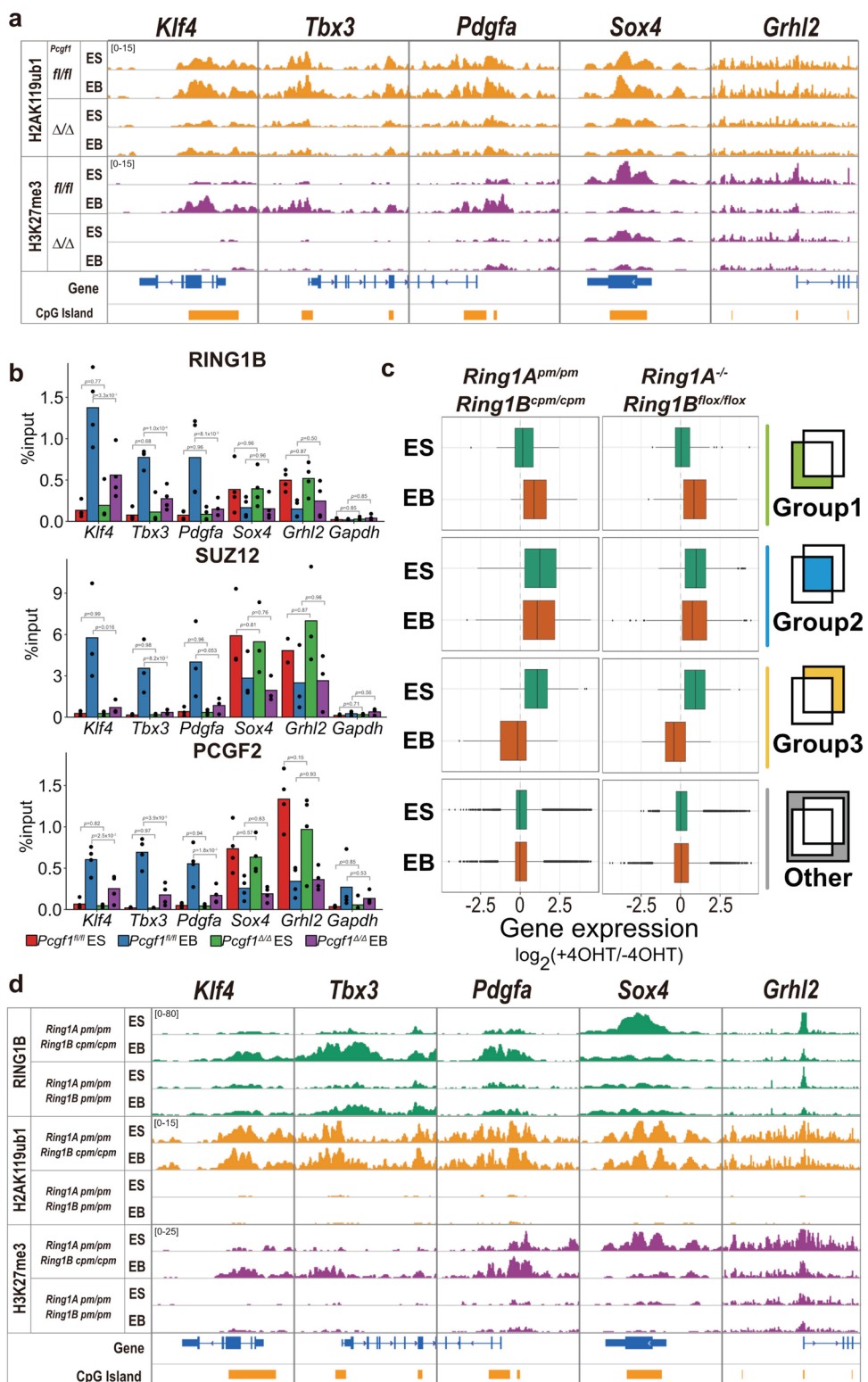

observation that recruitment of PRC2 and canonical PRC1 became evident only 8 h after differentiation induction (Supplementary Fig. 2i). PRC2 recruitment at target genes, therefore, could be preceded by downregulation of transcription. Indeed, inhibition of transcription per se has been reported to induce PRC2 recruitment in mouse ESC[3]. To directly test this model, we perturbed transcription by using Triptolide, which induces proteasomal degradation of RNA polymerase II (RNAPII). Twelve-

hour Triptolide treatment disrupted the binding of RNAPII to target TSSs, as expected (Supplementary Fig. 4a). As reported previously, we observed that SUZ12 was newly recruited to a group of genes in association with RING1B recruitment (Fig. 4a, Supplementary Fig. 4b). We compared the binding of RING1B and SUZ12 in Triptolide-treated or -untreated ESCs in differentially expressed genes clustered (C1 to C6) into six groups (Fig. 4a, Supplementary Fig. 4b; Supplementary Data 3). In

**Fig. 3 PCGF1 mediates H2AK119ub1-dependent recruitment of PRC2 upon differentiation. a** Deposition of H2AK119ub1 and H3K27me3 at selected Group 1 genes (*Klf4*, *Tbx3*, *Pdgfa*) or Group 3 genes (*Sox4*, *Grhl2*) in WT or *Pcgf1*-KO ESCs and EBs. Gene structures and positions of CpG islands are indicated at the bottom. **b** Binding of RING1B, SUZ12, and PCGF2 at selected Group 1 genes (*Klf4*, *Tbx3*, *Pdgfa*) and Group 3 genes (*Sox4*, *Grhl2*) in WT or *Pcgf1*-KO ESCs and EBs. Bar graphs of ChIP-qPCR results for RING1B, SUZ12, and PCGF2 in WT (*Pcgf1*fl/fl) or *Pcgf1*-KO (*Pcgf1*Δ/Δ) ESCs and EBs are shown. Error bars represent SEM ($n = 3$). *p*-value shows the significant differences calculated by Student's *t* test (one-sided). **c** Gene expression changes in each group (Group1: $n = 164$, Group2: $n = 1542$, Group3: $n = 73$) in *Ring1A/B* point mutant (*Ring1A*:pm/pm*Ring1B*:cpm/cpmERT2-CRE) ESCs and EBs. Box plot comparisons of fold changes in mRNA expression for each group in untreated (-4OHT) or *Ring1A/B* point mutant (+4OHT) ESCs and EBs are shown on the left. Gene expression changes in each group in *Ring1A/B*-dKO (+4OHT) or untreated (-4OHT) ESCs and EBs are shown as reference values in the right. The interquartile range (IQR) is defined as follow: the first quartile (Q1) is greater than 25% of the data and less than the other 75%. The second quartile (Q2) sits in the middle, dividing the data in half. Q2 is also known as the median. The third quartile (Q3) is larger than 75% of the data and smaller than the remaining 25%. In a box and whiskers plot, the ends of the box and its centerline mark the locations of these three quartiles. Each whisker extends to the furthest data point in each wing that is within 1.5 times the IQR. The outlier value is > 1.5 times and < 3 times the IQR beyond either end of the box. **d** Distribution of RING1B, H2AK119ub1, and H3K27me3 at selected Group 1 gene (*Klf4*, *Tbx3*, *Pdgfa*) and Group 3 genes (*Sox4*, *Grhl2*) in untreated or *Ring1* point mutant ESCs and EBs. Genomic snapshots from RING1B ChIP-Seq analysis, and H3K27me3 or H2AK119ub CUT&Tag analyses, are shown for untreated or *Ring1* point mutant ESCs and EBs.

untreated ESCs, SUZ12 and RING1B strongly bound genes in C1, C2, and C3; but not C4, C5, or C6. Genes in C4 and C5 were abundantly expressed and bound by RNAPII, compared to C1, C2, and C3 (Fig. 4b, Supplementary Fig. 4a). Consistent with this finding, Triptolide efficiently disrupted RNAPII binding at genes in C4 and C5, compared to C1, C2, and C3. Importantly, C4 genes showed a gain of SUZ12 binding upon Triptolide treatment, as previously reported[3] (Fig. 4a, c, Supplementary Fig. 4b, c). C4 genes also showed a gain of RING1B binding upon Triptolide treatment (Fig. 4a, c, Supplementary Fig. 4b). Of note, C4 genes involved 20% of Group 1 genes identified in Fig. 1b (32 out of 164 genes, including *Klf4*, *Tbx3*, and *Pdgfa*) (Supplementary Fig. 4d). Visual inspection at *Klf4*, *Tbx3*, and *Pdgfra* genes revealed Triptolide-induced accumulation of SUZ12 and RING1B (Fig. 4d). Therefore, inhibition of transcriptional activity induced recruitment of SUZ12 and RING1B to C4 genes. A similar tendency was also seen in C3 genes, which were more abundantly expressed than C1 and C2 genes, and exhibited more binding of SUZ12 and RING1B than C4 genes (Fig. 4b, Supplementary Fig. 4b). These findings indicate that following transcriptional inactivation, PRC2 and PRC1 are recruited to new target genes.

We asked whether SUZ12 and RING1B binding to C4 and C3 upon Triptolide treatment was mediated by PCGF1. Indeed, triptolide-induced accumulation of SUZ12 and RING1B at C4 and C3 genes were disrupted in the *Pcgf1*-KO (Fig. 4a, c, Supplementary Fig. 4b). Consistent with the role of H2AK119ub1 to recruit PRC2 to new target genes, we found that H2AK119ub1 accumulation in both C4 and C3 genes was disrupted in *Pcgf1*-KO (Fig. 4a). We confirmed this finding by visual inspection of SUZ12 and RING1B binding levels at *Klf4*, *Tbx3*, and *Pdgfa* genes (Fig. 4d). Based on these results, we propose that downregulation of transcriptional activity at the promoter triggers H2AK119ub1-mediated accumulation of PRC2, and canonical PRC1, at target genes.

To rule out potential off-target effects by Triptolide, we developed a drug inducible reporter gene system. This reporter possessed a human *Klf4* CGI, a Doxycycline (Dox)-inducible promoter harboring tetracycline responsive elements (TRE), and a downstream reporter EGFP gene (Fig. 5a). We established *Pcgf1* conditional ESCs with stable integration of this reporter. To drive transcription from the Dox-inducible promoter, we stably transfected a tetracycline trans-activator (*tTA*) gene and confirmed that EGFP was expressed in a Dox-dependent manner (Fig. 5b). H3K27 acetylation (H3K27ac), a mark of active transcription, was also induced in a Dox-dependent fashion (Supplementary Fig. 5a). We asked if the chromatin status of the reporter CGI could be regulated during the active-to-inactive transition of EGFP expression. Upon induced transcriptional

silencing by depletion of Dox, we detected accumulation of H3K27me3, RING1B, SUZ12, and PCGF2. KDM2B, exogenous PCGF1, and H2AK119ub1 accumulation were observed to a lesser extent (Fig. 5c, Supplementary Fig. 5b, c, d).

Finally, we asked whether the establishment of the PcG repressive domain in the Dox-inducible reporter system required PCGF1. Upon Dox removal, H2AK119ub1 and H3K27me3 deposition were reduced to about half of the control cells, in *Pcgf1*-KO cells (Figs. 5c, d, Supplementary Fig. 5d). Consistent with this result, we found that RING1B, SUZ12, and PCGF2 failed to accumulate at the reporter gene in the absence of PCGF1 (Fig. 5c). In contrast, H3K27ac levels did not change in the presence or absence of *Pcgf1* (Supplementary Fig. 5a). These findings clearly show that PCGF1 mediates the establishment of PcG-repressive domains at CGIs upon transcriptional inactivation.

## Discussion
**H2AK119ub1-dependent repressive pathway counteracts active transcription.** Our study reveals that a group of PcG target genes is progressively downregulated during differentiation and that a variant PRC1 complex harboring PCGF1 mediates differentiation-associated transition of transcriptional status at these genes. At the molecular level, PCGF1 facilitates deposition of H2AK119ub1 by forming a variant PCGF1-PRC1 complex at CGIs, which leads to H2AK119ub1-dependent consolidation of target gene silencing and recruitment of PRC2, and in turn canonical PRC1 recruitment through recognition of PRC2-mediated H3K27me3[14,20]. H2AK119ub1 deposition by PCGF1-PRC1 is, therefore, a prerequisite for PRC2 recruitment triggered by prior transcriptional downregulation, as reported previously[3]. In the transcriptionally active status, recognition of H2AK119ub1 by PRC2, likely PRC2.2, via JARID2 and AEBP2 could be restricted by ongoing transcriptional activity (Fig. 5d)[12]. However, it is still controversial how PCGF1-PRC1-mediated H2AK119ub1 contributes to the consolidation of gene silencing as earlier studies revealed that removal of either PRC2 or canonical PRC1 in ESCs has little impact on gene repression[14,20]. Intriguingly, H2AK119ub1 has been reported to directly disrupt the activity of RNAPII, as indicated by biochemical experiments[24–27]. H2AK119ub1 may, therefore, antagonize the transcriptional machinery once RNAPII activity is downregulated upon developmental signals.

Furthermore, our results indicate that PCGF1-mediated mechanisms could play a role in the de novo establishment of PcG-repressive domains at developmental genes, but not for the steady-state downregulation of PcG-target genes as observed in ESCs[20] and epiSCs (Supplementary Fig. 2k, l). Based on these observations, we

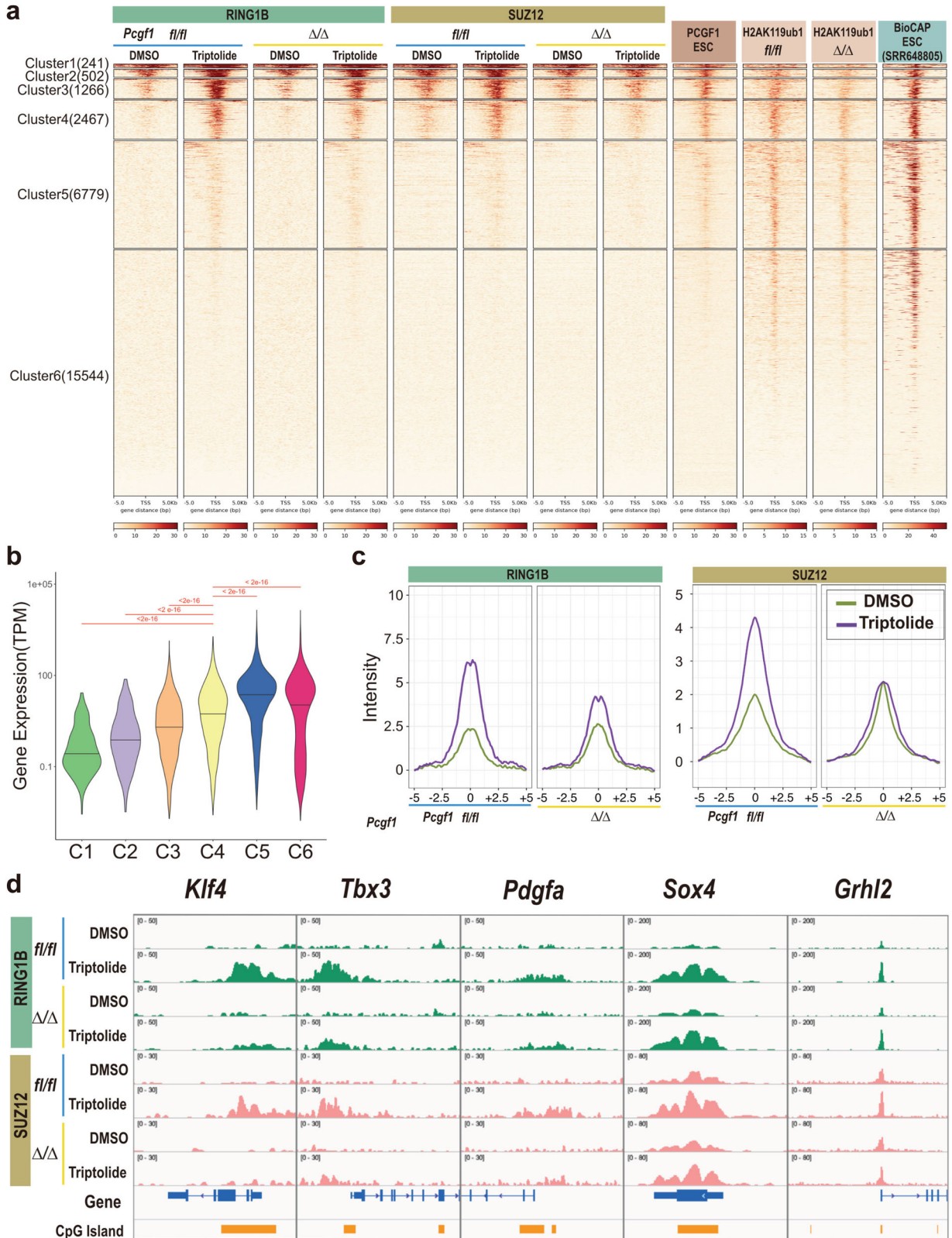

propose a role of PCGF1-PRC1 for prospective downregulation to transcriptionally active target genes associated with CGIs. Consistent with this role of PCGF1 in development, *Pcgf1*-KO exhibits disruption of differentiation abilities in ESCs, and a variety of other morphological defects, leading to early gestational lethality

(Supplementary Fig. 5e). Intriguingly, KDM2B, which is another core component of PCGF1-PRC1, has been reported to protect CGIs from ectopic DNA methylation[28]. The role of PCGF1-PRC1 to mediate the efficient transition of target CGI promoters from RNAPII-bound active status to PcG-repressed status may play a

**Fig. 4 Inhibition of RNA polymerase II facilitates PRC2 accumulation in a group of genes in a PCGF1-dependent manner. a** Heatmap views for RING1B and SUZ12 distribution at the TSS (5 kb upstream to 5 kb downstream) of each gene in WT (*fl/fl*) or *Pcgf1*-KO (*Δ/Δ*) ESCs, with or without (DMSO) Triptolide treatment. In parallel, the distribution of PCGF1 in untreated control ESCs, and H2AK119ub1 enrichment in control (*fl/fl*) or *Pcgf1*-KO (*Δ/Δ*) ESCs without Triptolide treatment, are shown as a reference. BioCAP data (SRR648805 [https://www.ncbi.nlm.nih.gov/geo/query/acc.cgi? acc=GSM1064680], deposited in GEO) showing the distribution of unmethylated CpGs are also shown. Genes were sorted into six groups by k-means clustering (*n* = 6) based on the distribution of RING1B, SUZ12, and PCGF1 in WT (*fl/fl*) ESCs. The number of genes in each cluster is shown within parentheses. **b** Violin plot showing differential gene expression in each cluster in untreated ESCs. Significant differences were calculated by Mann–Whitney *U* test (one-sided). The *p*-values are adjusted by BH methods. **c** Average binding of RING1B and SUZ12 in Cluster 4 genes in WT or *Pcgf1*-KO ESCs and EBs. Metaplot views of RING1B and SUZ12 binding at the TSS (5 kb upstream to 5 kb downstream) of Cluster 4 genes, in WTl (*fl/fl*) or *Pcgf1*-KO (*Δ/Δ*) ESCs, with or without triptolide treatment. Data for triptolide-treated, or untreated, ESCs are shown by purple and green lines, respectively. **d** Distribution of RING1B and SUZ12 at selected Group 1 genes (*Klf4*, *Tbx3*, *Pdgfa*) and Group 3 genes (*Sox4*, *Grhl2*) in control ESCs with or without (DMSO) Triptolide treatment. Genomic snapshots of RING1B and SUZ12 distributions in WT (*fl/fl*) or *Pcgf1*-KO (*Δ/Δ*) ESCs with or without (DMSO) triptolide treatment are shown.

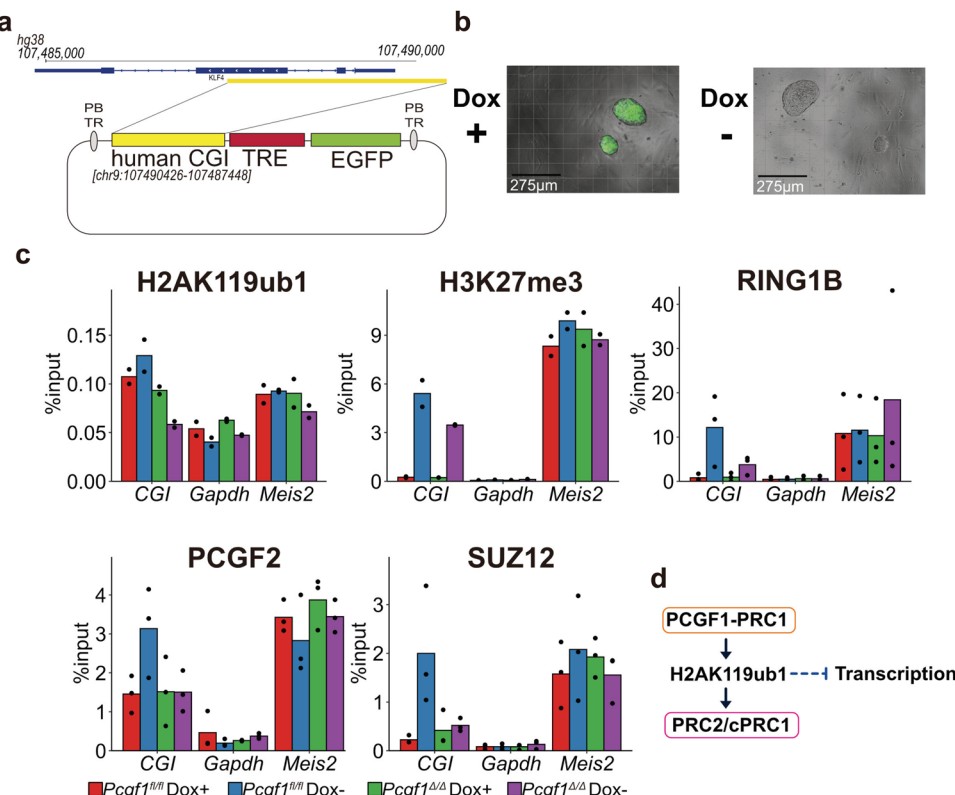

**Fig. 5 Transcriptional downregulation at *cis*-regulatory elements facilitates PRC2 recruitment to a CGI in a PCGF1-dependent manner. a** Schematic representation of the drug-inducible reporter construct. The vector contains a human *KLF4* CpG island (human CGI), a Dox-inducible promoter (TRE), and an EGFP gene. This reporter vector was stably integrated into *Pcgf1*-KO ESCs by piggyBAC transposase. **b** Dox-dependent expression of EGFP from the reporter construct in ESCs. Representative micrographs were shown. Experiments were repeated independently at least 2 times with the same results. **c** Dox- and PCGF1-dependent regulation of the accumulation of H2AK119ub1, H3K27me3, RING1B, SUZ12, and PCGF2 at the CGI of the reporter gene in ESCs. Bar graphs for ChIP-qPCR results for H2AK119ub1, H3K27me3, RING1B, SUZ12, and PCGF2 at the human CGI of the reporter, *Gapdh* and *Meis2* in WT (*Pcgf1*<sup>fl/fl</sup>) or *Pcgf1*-KO (*Pcgf1*<sup>Δ/Δ</sup>) ESCs in the absence or presence of Dox (-Dox and +Dox, respectively) are shown. Data are represented as mean ± SEM (*n* = 3). **d** Schematic summary for the role of PCGF1-PRC1 to link H2AK119ub1 with PRC2 revealed in this study. PCGF1-PRC1 mediates H2AK119ub1 deposition and contributes to the consolidation of gene silencing via undefined mechanisms (indicated by dotted line) and in turn recruitment of PRC2 and cPRC1 (indicated by an arrow). H2AK119ub1-dependent recruitment of PRC2 and canonical PRC1 is inhibited by transcriptional activity (indicated by a solid line).

role to suppress ectopic accumulation of DNA methylation and heterochromatin-related mechanisms[29].

**Diversity of PRC1-mediated mechanisms to establish new PcG-repressive domains.** We have previously shown that a variant PRC1 complex, containing PCGF3 and PCGF5 (PCGF3/5-PRC1), contributes to the initiation of PcG-mediated transcriptional silencing. This is manifested by the role of PCGF3/5-PRC1 to trigger *Xist*-mediated PRC2 recruitment at the inactive X

chromosome[19]. The PCGF3/5-PRC1 complex, furthermore, mediates downregulation of the *Meis2* gene in the distal region of forelimb buds in a retinoic acid-dependent manner[30]. Here, PCGF3/5-PRC1 facilitates downregulation of the *Meis2* gene by competing with transcriptional activators such as retinoic acid receptor complexes. Therefore, PCGF3/5-PRC1 mediated down-regulation of target genes does not necessarily require prior transcriptional inactivation. This feature of PCGF3/5-PRC1 is clearly different from that of PCGF1-PRC1, as we show that

PCGF1-PRC1 mediates consolidation of target gene silencing and establishment of PcG-repressive domains downstream of transcriptional inactivation. Consistently, PCGF3/5- and PCGF1-PRC1 should mediate H2K119ub1 in a mutually distinct manner, the former facilitating pervasive H2K119ub1 deposition and the latter facilitating CGI-directed H2K119ub1 deposition[20,31]. Moreover, we also suggest a potential role of variant PCGF6-PRC1 in this process (Fig. 2a). We, therefore, speculate that PRC1 contributes to the downregulation of target genes via differential usage of distinct PRC1 subcomplexes, likely to adapt to diverse of developmental signals during embryogenesis. Importantly, our recent study also revealed synergy among variant PRC1 subcomplexes to mediate H2K119ub1 at CGIs in ESCs, and during pre-implantation development[20,32]. It is worthy to note that PCGF6 was shown to complement PCGF1-mediated H2AK119ub1 for zygotic deposition of H3K27me3 in pre-implantation embryos[32]. We expect that a similar synergy could operate to initiate and consolidate downregulation of target genes, as shown by partial disruption of RING1B recruitment to the C4 genes upon Triptolide treatment (Fig. 4c). Conversely, a canonical PRC1 incorporating PCGF2 has been reported to play a role in activating a group of genes during mesoderm-directed differentiation of ESCs[33]. Such functional complexity of PRC1 may enable a robust and efficient transition of transcriptional status of target genes to generate appropriate cellular responses vis-à-vis the tremendous complexity of developmental signals operating in the developing embryo.

## Methods

**Generation of ESCs**. $Ring1A^{:\Delta/\Delta}Ring1B^{:fl/fl}:Rosa26^{CreERT2tg/+}$ [18], $Pcgf1^{:fl/fl}:Rosa26^{CreERT2tg/+}$ [19], $Pcgf2^{:fl/fl}:Pcgf4^{:fl/fl}:Rosa26^{CreERT2tg/+}$ [20], $Pcgf3^{:fl/fl}:Pcgf5^{:fl/fl}:Rosa26^{CreERT2tg/+}$ [19], and $Pcgf6^{:fl/fl}:Rosa26^{CreERT2tg/+}$ [21] were derived from respective blastocysts. Generation of $Ring1A^{:pm/pm}:Ring1B^{cpm/cpm}$ [16] ESCs were described previously. $Eed^{:fl/fl}:Rosa26^{CreERT2tg/+}$ ESCs were generated from mutant blastocysts according to the procedure described in our previous report[18].

**Mice**. $Eed^{fl}$ allele, in which exon 3 of $Eed$ can be conditionally deleted, was generated by using the targeting vector shown in Supplementary Fig. 3d. The targeting vector was introduced into ESCs by electroporation and homologous recombinants were identified by PCR-based screening. Germline chimeras were generated by aggregating morula stage embryos with homologous recombinant ESCs[21]. $Pcgf1^{fl/fl}$ mice were mated with $Pcgf1^{:+/+}CAG-Cre$ to generate $Pcgf1^{\Delta/+}$ mice. $Pcgf1^{\Delta/+}$ mice were intercrossed to generate $Pcgf1^{\Delta/\Delta}$ fetuses. All animal experiments and gene recombination experiments were performed according to the in-house guidelines for the care and use of laboratory animals [Approval number: 2021-011 and 2019-005(2)] and the in-house guidelines for the genetic recombination experiments (Approval number: Kei-29-011), respectively, of the RIKEN Center for Integrative Medical Sciences, Yokohama, Japan.

**Cell culture**. Embryonic stem cells (ESCs) were cultured in Dulbecco's minimum essential medium (DMEM) supplemented with 20% FBS, LIF (1000 U/mL), 2 μM SU5402 (Altair corporation), 0.8 μM PD184352 (Altair corporation), and 4 μM CHIR99021 (Altair corporation) on a feeder layer of mouse embryonic fibroblasts (MEFs). For embryoid body (EB) formation, ESCs were cultured without LIF, SU5402, PD184352, or CHIR99021 for 2 days on low attachment culture dishes. For epiblast stem cell (epiSC) establishment, EBs (day 2) were dispersed and cultured on MEFs with 20 ng/mL ActivinA (Peprotech), 12 ng/mL bFGF (Wako), and 2 μM IWP-2 (Wako) on MEFs in F12 medium (Gibco) containing 1% KSR (Gibco). To induce conditional deletion of respective genes, we added 0.8 μM 4-hydroxytamoxifen (4-OHT) into the culture medium according to the culture schedule shown in Supplementary Fig. 2b. Epiblast-like cells (EpiLCs) were induced by plating 100 K ESCs on human plasma fibronectin (16.7 μg/mL) coated 12-well plate in N2B27 basal medium supplemented with Activin A (20 ng/mL; Peprotech), bFGF (12 ng/mL; Invitrogen), and 1% KSR (Invitrogen) for 2 days[22]. Meanwhile, the medium was changed every day. For inhibition of RNA polymerase II (RNAPII), we treated ESCs with 10 μM Triptolide (Sigma) for 12 h.

**Gene expression analysis**. Total RNA was extracted using simplyRNA Tissue Kit (Promega). For RT-PCR analysis, cDNA was synthesized with the SuperScript™ VILO™ Master Mix (Thermo). RT-qPCR (Reverse Transcription Quantitative real-time PCR) was performed by using FastStart Universal SYBR Green Master (Roche). The expression of each gene was calibrated to $Gapdh$ gene expression as a housekeeping gene. For RNA-seq studies, mRNA was isolated from total RNA by using NEBNext Poly(A) mRNA magnetic isolation module (NEB), and sequencing libraries were generated according to the NEBNext Ultra II RNA library prep kit for illumina (NEB) protocol. RT-qPCR primers are listed in Supplementary Table 1.

**Chromatin immuno-precipitation (ChIP)**. For ChIP analysis, $10^6$–$10^7$ cells were cross-linked with 1% formaldehyde for 10 min at room temperature. Cross-linked cells were washed once with PBS, swelling buffer (0.1% NP-40, 1 mM DTT in PBS) was added, and samples were chilled on ice for 10 min. After centrifugation, pellets were suspended with RIPA buffer (10 mM Tris-HCl pH8.0, 1 mM EDTA, 140 mM NaCl, 1% Triton X-100, 0.1% SDS, 0.1% Sodium Deoxycholate) containing proteinase inhibitor (Roche, 04693124001), and then were sonicated by using a BioRuptor sonicator (Diagenode). After sonication, DNA concentration was measured using a Qubit 1× dsDNA HS Assay Kit (Thermo, cat. Q33230). Protein A/G beads were prepared as washed twice with bead blocking buffer and suspended with bead blocking buffer with approximately 5 μg of antibody and incubated at 4 °C for more than 2 h. Then, antibody-attached beads were washed twice with bead blocking buffer and the chromatin samples were added to the antibody-attached beads. After overnight incubation at 4 °C, chromatin sample-bound beads were washed 6 times with RIPA buffer, RIPA high salt buffer (10 mM Tris-HCl pH8.0, 1 mM EDTA, 500 mM NaCl, 1% Triton X-100, 0.1% SDS, 0.1% Sodium Deoxycholate) twice, LiCl buffer (10 mM Tris-HCl pH8.0, 1 mM EDTA, 250 mM LiCl, 0.5% NP-40, 0.5% Sodium Deoxycholate) twice and TE buffer twice. The bound materials were eluted with elution buffer (10 mM Tris-HCl pH8.0, 5 mM EDTA, 300 mM NaCl, 0.1% SDS) overnight at 65 °C and treated with RNaseA for 30 min at 37 °C and further incubated with Proteinase K for 1 h at 37 °C. Then DNA samples were purified and used for further ChIP-qPCR or ChIP-seq. ChIP-seq libraries were prepared by using the NEBNext Ultra II FS DNA prep kit (NEB E7805) following the manufacturer's instructions. ChIP-qPCR primers are listed in Supplementary Table 1. Antibodies are listed in Supplementary Table 2.

**CUT&Tag**. CUT&Tag was performed as described in Kaya-Okur et al.[34], with minor changes. pAG-Tn5 was used instead of pA-Tn5. pAG-Tn5 was constructed from pAG-MNase (addgene #123461) and pA-Tn5 (addgene #124601) plasmid DNAs. The pAG sequence was excised from the pAG-MNase plasmid by digestion with EcoRI-HindIII. The pAG fragment was integrated into the Tn5 vector from which the pA sequence was removed. Protein purification was performed as described in Kaya-Okur et al. ESCs and EB from either Pcgf1-KO or Ring1A/B mutants were dissociated with Accumax (Innovative Cell Technologies). For each assay, 100,000 cells were used with 10,000 HEK293T cells as spike-in control. Antibodies are listed in Supplementary Table 2.

**Protein extraction and Western blotting**. ESCs and EB were lysed with NETN buffer (50 mM Tris-HCl [pH 7.9], 500 mM NaCl, 1% NP-40, 1 mM EDTA [pH 8.0]) supplemented with cOmplete(Roche). Protein amount was quantified using Bio-Rad Protein Assay (Bio-rad). In total, 25 μg total protein was separated by SDS-PAGE and transferred to PVDF membrane using Trans-Blot Turbo Transfer System (Bio-Rad). Transfer membranes were blocked in 5% skim milk (Nakalai Tesque). Antibodies are listed in Supplementary Table 2.

**Construction of vectors**. For PCGF1-3TY1 expression, full-length Pcgf1 was PCR-amplified with a 3×TY1 epitope sequence at the N-terminus and inserted into the pCAG-IRES-Puro vector. To generate the PB-hKLF4_CGI-TRE3G-EGFP vector, a human KLF4 CGI was PCR-amplified from human genomic DNA and inserted upstream of the TRE3G promoter of PB-TRE3G-EGFP, and EGFP from pAcr3-EGFP was inserted downstream of the TRE3G promoter. Primers used for PCR were listed in Supplementary Table 1.

**Transfection**. For PCGF1-3TY1 expression, 1 μg of vectors were transfected into $10^5$ $Pcgf1^{:fl/fl}:Rosa26^{CreERT2tg/+}$ ESCs by using ViaFect™ Transfection Reagent (Promega, E4981). The cells were cultured with puromycin (1 μg/mL) for selection. The drug-inducible reporter system was generated by transfection. In total, 1 μg of PB-hKLF4_CGI-TRE3G-EGFP, 1 μg of pCAG-hyPBase, and PB-(CAG-Tet3G; EOS-C(3 + )-EGFP-IRES-puro)[35] were transfected into $10^5$ $Pcgf1^{:fl/fl}:Rosa26^{CreERT2tg/+}$ ESCs by using the ViaFect™ Transfection Reagent. The cells were cultured with G418 (200 μg/mL) and Doxycycline (1 μg/mL) for selection.

**Drug-inducible reporter system**. ESCs harboring the drug-inducible reporter system were cultured in ESC medium with Doxycycline (1 μg/mL). For promoter inactivation, Doxycycline was removed from culture for 2 days. For Pcgf1 ablation, ESCs were cultured for 2 days with 4-OHT and then cultured without Doxycycline for 2 days to induce promoter inactivation.

## Data analysis

*RNA-Seq*. Sequencing was performed with 75 bp single-end reads by NextSeq500 (run by Kazusa DNA Research Institute) or 50 bp single-end read by HiSeq1500 (run in RIKEN-IMS NGS facility). The data were mapped onto UCSC-mm9 from

Illumina iGenomes and counted the reads on exon using STAR (version 2.5.2b:–outSAMtype BAM SortedByCoordinate). Each gene reads on the exon was normalized by DESeq2 (version 1.22.1) for fold change and *p*-value calculation.

*ChIP-Seq.* Sequencing was performed with 75 bp single-end reads by NextSeq500 (run by Kazusa DNA Research Institute) or 50 bp single-end read by HiSeq1500 (run in RIKEN-IMS NGS facility). The data were mapped onto UCSC-mm9 from Illumina iGenomes. Generated sam format files were combined to bam format files using samtools view (version 1.3.1). Then, PCR duplicates and multiple aligned reads are removed. For bigwig generation, bam format files were processed by deeptools (version 2.5.0.1: bamCoverage). The data of CGI and TSS are derived from UCSC mm9.

*Determination of RING1B target genes.* To identify RING1B binding genes, we used ChIP-seq data for RING1B of *Pcgf* $^{fl/fl}$ ESCs and its derived EB. TSS ± 5 kb counts were done by BEDtools (version 2.26.0: intersect). RING1B target genes were defined based on a Gaussian Mixture Model using EM (expectation-maximization) algorism. The 99 percentiles of TSS counts were classified into two groups (mclust: mclustBIC modelNames = "V" G = 2). The threshold for RING1B targets was set over the score of 1 percentile of the second components. The gene list is shown in Supplemental file 1. Gene ontology (GO) term enrichment analyses were performed using DAVID (The Database for Annotation, Visualization and Integrated Discovery, https://david.ncifcrf.gov/).

*Meta plot around TSS for ChIP-Seq.* Mapped reads around TSS were counted by BEDtools (version 2.26.0: intersect). Calculation of correlation was done by HOMER annotatePeaks.pl (v4.11 -hist 100).

*Heatmap.* Mapped reads around TSS were counted by BEDtools (version 2.26.0: intersect). The top 25 genes were selected by the order of intensity calculated from the *Pcgf1* $^{fl/fl}$ EB data.

*Clustering and visualization of the Triptolide effect.* The read counts around TSS were done by deeptools (version 2.5.0.1: computeMatrix reference-point) using ChIP-Seq data for RING1B, SUZ12, PCGF1 obtained for *Pcgf* $^{fl/fl}$ ESCs.

*CUT&Tag.* Sequencing was performed with 50 × 2 bp pair-end reads by Nova-Seq6000 or 75 × 2 bp pair-end reads by NextSeq500. Paired-end fastq data were aligned to UCSC mm9-hg38 reference genome by Bowtie2 (version 2.3.2: default option). Generated sam format files were combined to bam format files using samtools view (version 1.3.1). PCR duplicate and multiple aligned reads are removed. For bigwig generation, bam format files were processed by deeptools (version 2.5.0.1: bamCoverage–scaleFactor). The spike-in scaling factor was calculated by the ratio of each human mapped reads and total reads.

*Meta plots around TSS for CUT&Tag.* Mapped reads around TSS were counted by BEDtools (version 2.26.0: intersect). Calculation of correlation was done by HOMER annotatePeaks.pl (v4.11 -hist 100). The spike-in scaling factor was calculated by the ratio of each human mapped reads and total reads.

## Data availability

The data that support this study are available from the corresponding author upon reasonable request. Raw and processed sequencing data generated in the course of this study can be accessed via the GEO database with accession number GSE141488. The ChIP-Seq data of triptolide-treated ESCs (Riising et al.[3]) are available under the accession numbers SRR1300952 and SRR1300956. The BioCAP data are available under the accession number SRR648805. Source data are provided with this paper.

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

## Acknowledgements

We thank Dr. Azusa Inoue (RIKEN) for the helpful discussion, Dr. Yukiko Gotoh (The University of Tokyo) for valuable comments, and Drs. Kyoichi Isono and Toshitada Takemori generously providing antibodies. This work was supported by RIKEN Junior Research Associate Program, Grant-in-Aid for Scientific Research from the Ministry of Education, Culture, Sports, Science and Technology of Japan (23249015 to H.K.; 19K07250), Grant-in-Aid for Scientific Research on Innovative Areas (JP19H05745 to H. K.), and by the Japan Agency for Medical Research and Development (AMED-CREST, 13417643 to H.K., T.K., S.I.). Work in the Klose lab is supported by the Wellcome Trust (209400/Z/17/Z) and the European Research Council (681440).

## Author contributions

Conceptualization: H.S., T.K., and H.K. Investigation: H.S., S.I., N.Y.-K., and Y.K. Bioinformatics: H.S. and E.K. Resources: S.I., Y.O., M.N., Y.K., N.P.B., and R.J.K. Writing: H.S. and H.K. Writing—review & editing: S.I., T.K., J.S., N.P.B., and R.J.K. Supervision: H.K. Funding acquisition: H.K.

## Competing interests

The authors declare no competing interests.
