## [Peer Review File · Nature Communications]

Reviewers' comments:

Reviewer #1 (Remarks to the Author):

Polycomb repressive complexes (PRC) are essential for maintaining cellular identity during development and can be divided to two major classes, PRC1 and PRC2. This manuscript explores the roles and molecular mechanisms of the Pcgf1-PRC1 complex during mouse embryonic stem cell (ESC) differentiation to embryoid bodies (EBs). The authors generated comprehensive genomic data, including RNA-seq, ChIP-seq and CUT&Tag-seq, in series of conditional knock-out ESCs of various Pcgf proteins, both before and after induction to differentiate to EBs. Their main finding is that the Pcgf1-PRC1 complex is recruited to Polycomb target genes and then this leads to deposition of H2AK119ub1 and subsequent deposition of PRC2. The experiments are very well executed and the work is clearly explained. This will be of broad interest to those in the chromatin and stem cell fields who are interested in the regulation of differentiation by Polycomb proteins. My comments below are minor suggestions to further help and clarify some aspects.

Minor points:

1. In the introduction, the authors should update to also mention recent work that established that PRC2.1 and PRC2.2 are both required for deposition of H3K27me3 (Helin and Bracken labs, 2019) and that variant PRC1 mediated H2AK119ub1, promotes primarily PRC2.2 mediated deposition of H3K27me3 (Klose and Pasini labs, 2020).
2. For Figure 1a and 1b, the authors should state that what type of ESCs were used for the ESC-EB differentiation.
3. The authors should show gene ontology enrichment analysis of the three groups of genes in Figure 1b.
4. In the results section, the authors should clarify whether their statement "The same trend was also seen, albeit in a weaker level, in Group 2 and 3 genes (Extended Fig. 2a)". relates to Figure 2a and not Extended Figure 2a.
5. For Figure 4a and 4b, the authors could perform a western blot analysis of Pcgf1 to confirm it was fully depleted upon Dox induction.

Reviewer #2 (Remarks to the Author):

In this manuscript, Koseki and colleagues interrogated how PRC1 complexes regulate gene expression during embryonic stem cell (ESC) differentiation into embryoid bodies (EBs). By using gene expression profiles, ChIP-seq experiments, and KO cells each of the six PCGF orthologs, they found that PCGF1 was required for initial downregulation of a sub-set genes in two days-old EBs. Also, they suggest that PCGF1 was required for efficient RING1B, PRC2 and canonical PRC1 complexes during differentiation in this order of events. Albeit I believe that two day-old EBs are quite heterogeneous and therefore I would have chosen another differentiation model (i.e ESCs to EpiSCs, 2i +LIF to -LIF), their conclusions are well supported by the results. I do have the following specific comments which I hope will improve the impact of the study:

1. How many genes were deregulated in the EBs compared to the ESC? Also, I think it would be informative to include a 24h time point in both the RNA-seq experiments in WT cells. Moreover, given the fact that the 8h time point seems to be quite important for the interpretation of the results, occupancy of PCGF1, RING1B, PCGF2 and PRC2 should be assessed by ChIP-qPCR in differentiation time course (0, 8h, 24 and 48h). Analysis of KLF4, TBX3, and PDGFA genes should be enough. These experiments will confirm the recruitment events proposed in Figures 3 and 4.
2. Group 1 genes were affected in PCGF6 KO cells, thus authors should determine if PCGF6 binds to this group of genes in ESCs and EBs. Could it be possible to generate PCGF1/6 dKO cells? These experiments will elucidate whether both PCGF orthologs functionally cooperate in regulating Group 1 genes.

3. I was surprised that PCGF1 was not bound to Groups 1 to 3 in ESCs (Fig. 2a) and more specially to Group 2. These groups are based on RING1B occupancy in both ESC and EBs, and we know that PCGF1 is expressed and co-localizes with RING1B in ESCs. In line with this, I was also surprised with the low number of RING1B targets the authors found compared to published RING1B ChIP-seq in ESCs. The authors should clarify this. Moreover, H2AK119ub1 results from Figure 3a suggest that either PCGF1 was recruited to Group 1 genes in ESCs or that a PRC1 complex with PCGF6 deposited the mark.

4. If RING1B was not recruited to KLF4, TBX3, and PDGFA genes in ESCs but H2AK119ub1 was present, RING1A must be the E3-ligase involved in depositing H2AK119ub1 at these sites. Can the authors check this hypothesis by doing RING1A ChIP-qPCR?

5. The key control in Figure 3C should be to compare by RNA-seq the expression of Group 1 genes in control and RING1B mutant ESCs.

6. To functionally demonstrate that PCGF1 and PRC2 are required for full downregulation of Group 1 genes during ESC differentiation, authors should deplete EED (by shRNA) in PCGF1 KO cells.

Minor points:

1. PcG genes are regulated during the cell cycle. The authors should check the cell cycle profile of the cells during the differentiation time course presented in Figure S1C. If possible, PCGF orthologs expression by RT-qPCR should also be evaluated.

2. The authors should acknowledge the published work on PCGF2/MEL18 and PCGF4/BMI1 during ESC differentiation towards the mesoderm and ectoderm lineages, respectively.

3. "The same trend was also seen, albeit in a weaker level, in Group 2 and 3 genes (Extended Fig. 2a)." should read (Fig. 2a).

4. Are SOX4 and GRHL2 genes PCGF1 targets in ESCs?

5. I don't agree that KDM2B binding was unchanged in In Fig. S4, I think it was not recruited.

Reviewer #3 (Remarks to the Author):

In the manuscript entitled "Variant PCGF1-PRC1 links PRC2 recruitment with differentiation-associated transcriptional inactivation at target genes" the authors elucidate the contribution of distinct Polycomb activities in the downregulating the expression of target genes upon differentiation of mESC into EBs. The authors provide evidences that the PCGF1-containing PRC1 complex is crucial to facilitate differentiation-associated down-regulation of a specific group of genes upon transition from mESC to EBs. They suggest a model in which the PCGF1-vPRC1 complex is able to initiate PcG-repressive domain formation by recruiting RING1B to deposit H2AK119ub1 via CpG island recognition by KDM2B at target genes. This allows subsequent recruitment of PRC2, deposition of H3K27me3 followed by PCGF2-cPRC1 recruitment enforcing the PcG-mediated gene silencing.

Overall the data presented are of good quality and largely clear. The way some expression-location analyses are presented is sometime difficult to follow and revising their presentation may help to follow their experiments. Some data are also redundant and repeated in the figures (Fig.1d and Fig.3c), this should be avoided. The presented data and proposed model is rather confirmatory with limited novelty. However, the major claim of this manuscript is the central involvement of PCGF1 in the de novo establishment of PcG-repressive domains at developmental genes. This is in some part convincing, however, the manuscript also fell short in fully demonstrating the exclusive role of PCGF1 in mounting repression at targets during differentiation.

The use of EBs remains rather dishomogeneous as cell differentiation is not fully restricted to a single fate and the results could be affected by a diverse abundance in cell types. I do understand that the time at which EBs are taken is rather short but these results will be strengthened by also differentiating ESC into more

homogenous fates using alternative protocols (see also following comment).

The authors undermine the effects observed with the PCGF6 KO, however, it remains unclear why the same group of genes fail to be repressed in the absence also of PCGF6 (Fig.1d). Is PCGF6 involved in PCGF1 mediated silencing? If so, how does this reconcile with the central recruitment of H3K27me3 and canonical PRC1 shown later in the figures? The interplay between non-canonical PRC1 complexes must be better investigated in order to determine whether PCGF1 plays an apical role in de novo PcG repression or simply acts redundantly with other PRC1 complexes as previously suggested by the authors in ESC (Fursova et al. Mol Cell 2019). The authors cannot exclude that lack of repression is a secondary effect of differentiation failure of PCGF1 KOs in EBs (or PCGF6 KOs). De novo establishment of target repression with other stimuli is therefore required to reinforce their observations.

The authors also fail to determine how PCGF1 activity is regulated at physiological levels. Using ectopic expression of PCGF1 they show a nice recruitment at Group 1 genes in EBs (Fig.2a). However the same trend is also evident for Group 3 genes which undergo transcriptional activation in normal EBs (Fig.2b). This suggests that the observed PCGF1 recruitment is rather the consequence of overexpression conditions. Indeed, it also remains unclear why H2Aub deposition is not modulated in the presence of a clear recruitment of PRC1 activity at these genes. This deposition is not fully dependent on PCGF1 activity (Fig.3a) suggesting that multiple complexes are already active at these sites (Fig.3a). The only activity which is clearly modulated is the recruitment of PRC2 and canonical PRC1. This likely explains the strong recruitment of RING1B as was previously shown by others to be dependent on non-canonical PRC1 forms (Tavares et al. Cell 2012). Using catalytically dead RING mutants, the authors further show that deposition of H2Aub is required for PRC2 recruitment at de novo EBs targets (group 1). However, lack of modulation in H2Aub deposition does not fit with a central role of H2Aub in building de novo repressive PcG modules. The proposed model also does not fit with the central role of H2Aub in the repression of target genes published by the same authors earlier this year (Blackledge et al. Mol Cell 2020) as well as with the marginal role of PRC2 and canonical PRC1 complexes in transcriptional repression (Rising et al. Mol Cell 2014, Tamburri et al. Mol Cell 2020 and Fursova et al. Mol Cell 2020). How is it possible that H2Aub is required to repress these genes in EBs but is not differentially deposited?

The figure 4 is fine, however, it does not add much to the story and shows a poor modulation of PRC2 recruitment. The modulation of H3K27me3 and H2Aub at artificial CGI is also not convincing and it remains unclear while these modifications have not been tested with quantitative PCRs like the rest.

We would like to submit the revised version of our manuscript N-COMMS-20-04383 to *Nature Communications*. We greatly appreciate the comments and suggestions made by the reviewers. To address the reviewers' comments, we have newly added (i) Figure 4, (ii) Extended Figures 1 and 4, (iii) rearranged Figures 1 and 2, and, (iv) Extended Figures 2, 3 and 5. Because of the inclusion of these experiments, we revised and rewritten some parts of the manuscript. The newly added, or altered, sections in the text are shown in blue. Please find the point-by-point replies to each of the reviewers' comments below.

With the best wishes,

Haruhiko Koseki, MD, PhD.

Point-by-point replies to reviewers:

Reviewer #1:

Polycomb repressive complexes (PRC) are essential for maintaining cellular identity during development and can be divided to two major classes, PRC1 and PRC2. This manuscript explores the roles and molecular mechanisms of the Pcgf1-PRC1 complex during mouse embryonic stem cell (ESC) differentiation to embryoid bodies (EBs). The authors generated comprehensive genomic data, including RNA-seq, ChIP-seq and CUT&Tag-seq, in series of conditional knock-out ESCs of various Pcgf proteins, both before and after induction to differentiate to EBs. Their main finding is that the Pcgf1-PRC1 complex is recruited to Polycomb target genes and then this leads to deposition of H2AK119ub1 and subsequent deposition of PRC2. The experiments are very well executed and the work is clearly explained. This will be of broad interest to those in the chromatin and stem cell fields who are interested in the regulation of differentiation by Polycomb proteins. My comments below are minor suggestions to further help and clarify some aspects.

Minor points:

1. *In the introduction, the authors should update to also mention recent work that established that PRC2.1 and PRC2.2 are both required for deposition of H3K27me3 (Helin and Bracken labs, 2019) and that variant PRC1 mediated H2AK119ub1, promotes primarily PRC2.2 mediated deposition of H3K27me3 (Klose and Pasini labs, 2020).*

We thank the reviewer for pointing out these very relevant papers, which are now included in the introduction (page 2, lines 20-24) and discussion (page 10, lines 24-29).

2. *For Figure 1a and 1b, the authors should state that what type of ESCs were used for the ESC-EB differentiation.*

We have added the information on origin and culture conditions of ESCs in the result (page 3, lines 17-23), and materials and methods (page 18, line 24-page 19, line 6).

3. *The authors should show gene ontology enrichment analysis of the three groups of genes in Figure 1b.*

We appreciate the reviewer's suggestion. We have added the GO data in Extended Figure 1. Please also see the results section (page 3, lines 30-32).

4. *In the results section, the authors should clarify whether their statement "The same trend was also seen, albeit in a weaker level, in Group 2 and 3 genes (Extended Fig. 2a)". relates to Figure 2a and not Extended Figure 2a.*

We have corrected the text accordingly.

5. *For Figure 4a and 4b, the authors could perform a western blot analysis of Pcgf1 to confirm it was fully depleted upon Dox induction.*

The cell line used in our paper is the same one reported in Almeida et al. (2017) and Fursova et al. (2019). As both of those papers show western blot results, we did not perform this experiment again. We would also like to point out that in our experiment, Dox is used to activate TRE-mediated expression of EGFP, but not modulate PCGF1 expression.

Reviewer #2:

In this manuscript, Koseki and colleagues interrogated how PRC1 complexes regulate gene expression during embryonic stem cell (ESC) differentiation into embryoid bodies (EBs). By using gene expression profiles, ChIP-seq experiments, and KO cells each of the six PCGF orthologs, they found that PCGF1 was required for initial downregulation of a sub-set genes in two days-old EBs. Also, they suggest that PCGF1 was required for efficient RING1B, PRC2 and canonical PRC1 complexes during differentiation in this order of events.

Albeit I believe that two day-old EBs are quite heterogeneous and therefore I would have chosen another differentiation model (i.e ESCs to EpiSCs, 2i +LIF to -LIF), their conclusions are well supported by the results.

We appreciate the reviewer's concern about the culture conditions for induction of EBs. We maintained the ESCs in LIF and 3i, on feeder MEFs, and induced differentiation to EBs by depleting LIF, 3i, and feeder MEFs. In this condition, we observed that ESCs differentiate into EBs in a synchronous manner. The detailed description of the culture conditions is now added to the text (page 3, lines 17-23: page 18, line 24-page 19, line 6). In addition, we confirmed PCGF1-dependent accumulation of RING1B to Klf4 and Tbx3, by differentiating ESCs into epiblast like cells (epiLCs) (Extended Figures 2g, 2h) (page 5, lines 11-16).

I do have the following specific comments which I hope will improve the impact of the study:

1. How many genes were deregulated in the EBs compared to the ESC? Also, I think it would be informative to include a 24h time point in both the RNA-seq

experiments in WT cells. Moreover, given the fact that the 8h time point seems to be quite important for the interpretation of the results, occupancy of PCGF1, RING1B, PCGF2 and PRC2 should be assessed by ChIP-qPCR in differentiation time course (0, 8h, 24 and 48h). Analysis of KLF4, TBX3, and PDGFA genes should be enough. These experiments will confirm the recruitment events proposed in Figures 3 and 4.

Upon ESC-to-EB differentiation, we found 765 upregulated genes ($\log_2FC > 2$) and 625 downregulated genes ($\log_2FC < -2$). (page4, lines 2-4). We appreciate the suggestion to investigate bindings of RING1B, PCGF2 and SUZ12 during ESC-to-EB transition. We performed time course ChIP-qPCR analysis and observed progressive accumulation of RING1B, PCGF2 and SUZ12 after the 8-hour stage. These results are shown in Extended Figure 2j, and described in the result section (page 5, lines 25-29).

2. Group 1 genes were affected in PCGF6 KO cells, thus authors should determine if PCGF6 binds to this group of genes in ESCs and EBs. Could it be possible to generate PCGF1/6 dKO cells? These experiments will elucidate whether both PCGF orthologs functionally cooperate in regulating Group 1 genes.

We appreciate the reviewer's comment. Unfortunately, ChIP-grade antibody for PCGF6 is not available, and therefore we could not perform this experiment. Instead, we investigated RING1B distribution in *Pcgf6*-KO in ESCs and EBs, which we believe is the major issue here. We did not find noticeable changes in RING1B binding in *Pcgf6*-KO ESCs or EBs and, therefore, concluded that PCGF1 played a major role to bring RING1B to the Group 1 genes during ESC-to-EB differentiation. These results are shown in Extended Figures 2d and 2e, and described in the results section (page 4, line 31-page 5, line 7).

3. I was surprised that PCGF1 was not bound to Groups 1 to 3 in ESCs (Fig. 2a) and more specially to Group 2. These groups are based on RING1B occupancy in both ESC and EBs, and we know that PCGF1 is expressed and co-localizes with RING1B in ESCs. In line with this, I was also surprised with the low number

of RING1B targets the authors found compared to published RING1B ChIP-seq in ESCs. The authors should clarify this. Moreover, H2AK119ub1 results from Figure 3a suggest that either PCGF1 was recruited to Group 1 genes in ESCs or that a PRC1 complex with PCGF6 deposited the mark.

Upon the comments from the reviewer, we re-examined PCGF1 distribution by using anti-PCGF1 antibody raised by Fursova et al. (2019), and replaced TY1-tagged PCGF1 (expressed in *Pcgf1*-KO ESCs) ChIP-seq with endogenous PCGF1 ChIP-seq. We successfully detected binding of endogenous PCGF1 to target genes in both ESCs and EBs (not only Group 2 and 3 genes, but also Group 1 genes) by using the endogenous antibody. These results are shown in Figures 2b and 2c in the revised version and described in the results section (page 6, lines 13-25).

For RING1B, we identified 1615 and 1706 target genes in ESCs and EBs, respectively. In our previous study (Endoh et al., 2017, eLife), we identified 2959 genes bound by RING1B. Although the number of genes found in this study is smaller than our previous study, such differences are mainly caused by threshold setting. To support this notion, we observed most of the genes identified in this study were also included in our previous study (given the list is quite long, the comparison data is not shown in the manuscript).

4. If RING1B was not recruited to KLF4, TBX3, and PDGFA genes in ESCs but H2AK119ub1 was present, RING1A must be the E3-ligase involved in depositing H2AK119ub1 at these sites. Can the authors check this hypothesis by doing RING1A ChIP-qPCR?

We thank the reviewer for this comment. The short answer is that there is no ChIP-grade antibody for RING1A as far as we know. Instead, our results indicate that newly recruited RING1B to Group 1 genes upon ESC-to-EB differentiation (Figures 2b, 2c) mainly represents RING1B incorporated in canonical PRC1 rather than variant PCGF1-PRC1, as described in page 6, line 5-page 7, line 3. We indeed found substantial binding of PCGF1 and RING1B in Group 1 genes in ESCs (Figures 1c, 1d, and Figures 2b, 2c); which mediates H2AK119ub1, as

confirmed by deletion of *Pcgf1* (Figure 3a), and disruption of the catalytic activity of *Ring1A/B* (Figure 3d). Therefore, modest but clear binding of PCGF1-PRC1 is observed at Group 1 genes, which we believe contributes to H2AK119ub1 deposition (page 6, line 5-page 8, line 18).

5. *The key control in Figure 3C should be to compare by RNA-seq the expression of Group 1 genes in control and RING1B mutant ESCs.*

We appreciate the reviewer's comment. We have now modified Figure 3c to show gene expression changes in point mutant *Ring1A/B*, and *Ring1A/B* double KO, in parallel. These results are discussed in the results section (page 7, lines 27-page 8, line 6).

6. *To functionally demonstrate that PCGF1 and PRC2 are required for full downregulation of Group 1 genes during ESC differentiation, authors should deplete EED (by shRNA) in PCGF1 KO cells.*

We thank the reviewer for this comment. We agree that it is important to elucidate how PCGF1 and PRC2 synergistic downregulate target genes. In this study, we propose that PCGF1-dependent H2AK119ub1 recruits PRC2 (Figure 5d). To strengthen this model, we have genetically ablated PRC2 activity (*Eed*-KO) in ESCs and investigated PCGF1 distribution in ESCs and EBs. We did not find clear changes in PCGF1 distribution in *Eed*-KO ESCs or EBs. This supports our model that PCGF1 functions upstream of PRC2, and RING1B (incorporated in canonical PRC1 [cPRC1]). The results are shown in Extended Figures 3d-3h and described in the results section (page 8, lines 7-18).

Minor points:

1. *PcG genes are regulated during the cell cycle. The authors should check the cell cycle profile of the cells during the differentiation time course presented in Figure S1C. If possible, PCGF orthologs expression by RT-qPCR should also be evaluated.*

We have investigated proliferation of *Pcgf1*-KO during ESC-to-EB differentiation and observed moderate decrease of the proliferation rate. We have also examined the expression of *Pcgf* factors during during ESC-to-EB differentiation as shown in Extended Figure 2a (page 4, lines19-21).

2. *The authors should acknowledge the published work on PCGF2/MEL18 and PCGF4/BMI1 during ESC differentiation towards the mesoderm and ectoderm lineages, respectively.*

We thank the reviewer for pointing out this issue. We have included a paper by Morey et al (Cell stem cell, 2015) in the discussion section (page 12, lines 7-8).

3. *“The same trend was also seen, albeit in a weaker level, in Group 2 and 3 genes (Extended Fig. 2a).” should read (Fig. 2a).*

Thank you for this comment. We have corrected the text accordingly.

4. *Are SOX4 and GRHL2 genes PCGF1 targets in ESCs?*

Yes, this is shown in Figure 2c.

5. *I don't agree that KDM2B binding was unchanged in In Fig. S4, I think it was not recruited.*

Thank you for this comment. We have revised the text accordingly (Page 10, lines 4-5).

Reviewer #3 (Remarks to the Author):

In the manuscript entitled “Variant PCGF1-PRC1 links PRC2 recruitment with differentiation-associated transcriptional inactivation at target genes” the authors elucidate the contribution of distinct Polycomb activities in the downregulating the expression of target genes upon differentiation of mESC into EBs. The authors provide evidences that the PCGF1-containing PRC1 complex is crucial to facilitate differentiation-associated down-regulation of a specific group of genes upon transition from mESC to EBs. They suggest a model in which the PCGF1-vPRC1 complex is able to initiate PcG-repressive domain formation by recruiting RING1B to deposit H2AK119ub1 via CpG island recognition by KDM2B at target genes. This allows subsequent recruitment of PRC2, deposition of H3K27me3 followed by PCGF2-cPRC1 recruitment enforcing the PcG-mediated gene silencing.

1. Overall the data presented are of good quality and largely clear. The way some expression-location analyses are presented is sometime difficult to follow and revising their presentation may help to follow their experiments. Some data are also redundant and repeated in the figures (Fig.1d and Fig.3c), this should be avoided. The presented data and proposed model is rather confirmatory with limited novelty. However, the major claim of this manuscript is the central involvement of PCGF1 in the de novo establishment of PcG-repressive domains at developmental genes. This is in some part convincing, however, the manuscript also fell short in fully demonstrating the exclusive role of PCGF1 in mounting repression at targets during differentiation.

We regret the confusion in Figures 1d and 3c. Figure 3c showed the results for *Ring1A/B* point mutant ESCs, while Figure 1d showed the results for *Ring1A/B* double KO ESCs. Reviewer 2 (comment #5) suggested to show both results in parallel. We therefore revised Figure 3c as suggested by reviewer 2.

2. The use of EBs remains rather dishomogeneous as cell differentiation is not fully restricted to a single fate and the results could be affected by a diverse

abundance in cell types. I do understand that the time at which EBs are taken is rather short but these results will be strengthened by also differentiating ESC into more homogenous fates using alternative protocols (see also following comment).

We appreciate the reviewer's concern. A similar comment was also made by reviewer 2. We maintained the ESCs in LIF and 3i, on feeder MEFs, and induced differentiation to EBs by depleting LIF, 3i, and feeder MEFs. In this condition, we observed that ESCs differentiate into EBs in a synchronous manner. The detailed description of the culture conditions is now added to the text (page 3, lines 17-23: page 18, line 24-page 19, line 6). In addition, we confirmed PCGF1-dependent accumulation of RING1B to Klf4 and Tbx3, by differentiating ESCs into epiblast like cells (epiLCs) (Extended Fig. 2g, 2h) (page 5, lines 11-16).

3. *The authors undermine the effects observed with the PCGF6 KO, however, it remains unclear why the same group of genes fail to be repressed in the absence also of PCGF6 (Fig.1d). Is PCGF6 involved in PCGF1 mediated silencing? If so, how does this reconcile with the central recruitment of H3K27me3 and canonical PRC1 shown later in the figures? The interplay between non-canonical PRC1 complexes must be better investigated in order to determine whether PCGF1 plays an apical role in de novo PcG repression or simply acts redundantly with other PRC1 complexes as previously suggested by the authors in ESC (Fursova et al. Mol Cell 2019).*

We appreciate this comment. A similar concern was also raised by reviewer 2 (Comment #2). We therefore investigated if RING1B distribution was altered in *Pcgf6*-KO in ESCs and EBs. We did not find changes in RING1B binding in *Pcgf6*-KO ESCs or EBs and, therefore, concluded that PCGF1 played a major role to bring RING1B to the Group 1 genes during ESC-to-EB differentiation. These results are shown in Extended Fig. 2d and 2e and described in the results section (page 4, line 31-page 5, line 6). Our recent study, however, revealed that PCGF6 complemented PCGF1-mediated H2AK119ub1 during pre-implantation development. In this case, H2AK119ub1 mediated by PCGF1-PRC1 and

PCGF6-PRC1 both guide zygotic deposition of H3K27me3 (see discussion, page 11, line 33-page 12, line 4).

4. The authors cannot exclude that lack of repression is a secondary effect of differentiation failure of PCGF1 KOs in EBs (or PCGF6 KOs). De novo establishment of target repression with other stimuli is therefore required to reinforce their observations.

The idea that H2AK119ub1 silences gene expression, is still controversial to some extent, and needs further validation (see discussion, page 10, lines 16-33). However, we did note a role for H2AK119ub1 to recruit PRC2, and canonical PRC1, upon transcriptional down-regulation. This role is independent from differentiation of ESCs, and instead is dependent on prior transcriptional downregulation (see Figures 4 and 5).

5. The authors also fail to determine how PCGF1 activity is regulated at physiological levels. Using ectopic expression of PCGF1 they show a nice recruitment at Group 1 genes in EBs (Fig.2a). However the same trend is also evident for Group 3 genes which undergo transcriptional activation in normal EBs (Fig.2b). This suggest that the observed PCGF1 recruitment is rather the consequence of overexpression conditions. Indeed, it also remains unclear why H2Aub deposition is not modulated in the presence of a clear recruitment of PRC1 activity at these genes. This deposition is not fully dependent on PCGF1 activity (Fig.3a) suggesting that multiple complexes are already active at these sites (Fig.3a). The only activity which is clearly modulated is the recruitment of PRC2 and canonical PRC1. This likely explains the strong recruitment of RING1B as was previously shown by others to be dependent on non-canonical PRC1 forms (Tavares et al. Cell 2012). Using catalytically dead RING mutants, the authors further show that deposition of H2Aub is required for PRC2 recruitment at de novo EBs targets (group 1). However, lack of modulation in H2Aub deposition does not fit with a central role of H2Aub in building de novo repressive PcG modules. The proposed model also does not fit with the central role of H2Aub in the repression of target genes published by the same authors earlier

this year (Blackledge et al. Mol Cell 2020) as well as with the marginal role of PRC2 and canonical PRC1 complexes in transcriptional repression (Riising et al. Mol Cell 2014, Tamburri et al. Mol Cell 2020 and Fursova et al. Mol Cell 2020). How is it possible that H2Aub is required to repress these genes in EBs but is not differentially deposited?

To address this comment, we have re-examined PCGF1 distribution by using an anti-PCGF1 antibody raised by Fursova et al. (2019), and replaced Ty1-PCGF1 ChIP-seq data with endogenous PCGF1 ChIP-seq. We successfully detected binding of endogenous PCGF1 to target loci including not only Group 2 and 3 genes, but also Group 1 genes in ESCs (Fig. 2b, 2c). PCGF1, therefore, binds target CGIs in a constitutive manner; and mediates H2AK119ub1 at Group 1 genes, as well as Group 2 and 3 genes, in ESCs (Fig. 3a, Extended Fig. 3a). Importantly, we also found modest but clear binding of RING1B in Group 1 genes in ESCs (Fig. 1c, d, and Fig. 2b, c). This binding likely mediates H2AK119ub1, as represented by a lack of H2AK119ub1 depositions in RING1A/B catalytic mutant ESCs (Fig. 3d). We, therefore, conclude that a fraction of RING1B that forms complexes with PCGF1, and constitutively binds to the Group 1 genes, irrespective of the transcriptional status. This RING1B/PCGF1 fraction mediates H2AK119ub1 at the Group 1 genes in ESCs. In contrast, increase of RING1B binding at the Group 1 genes during ESC-to-EB transition represents RING1B incorporated in canonical PRC1, which is recruited to target genes via recognition of PRC2-mediated H3K27me3 (3a, 3b). We therefore suggest that PCGF1-PRC1 mediates H2AK119ub1 at not only transcriptionally silenced genes, but also active genes (such as Group 1 genes). We further reveal that at transcriptionally active genes, H2AK119ub1-dependent recruitment of PRC2 and canonical PRC1 is inhibited by transcriptional activity *per se* (Figures 4, 5d). We speculate that the H2AK119ub1-dependent repressive pathway counteracts transcriptional activity. Upon downregulation of transcription, gene silencing occurs via recruitment of PRC2 and canonical PRC1 (see page 8, line 20-page 11, line 13).

6. *The figure 4 is fine, however, it does not add much to the story and shows a poor modulation of PRC2 recruitment. The modulation of H3K27me3 and H2Aub*

at artificial CGI is also not convincing and it remains unclear while these modifications have not been tested with quantitative PCRs like the rest.

In Figure 4 of the revised version, we reveal the impact of prior transcriptional inactivation by Triptolide on H2AK119ub1-dependent recruitment of PRC2 and canonical PRC1 (Figure 4, Extended Figure 4). In Figure 5 (Figure 4 in the previous version), we addressed the same issue using a different approach to avoid potential off-target effects by Triptolide. These results were further validated by CHIP-qPCR (Figure 5c, Extended Figures 5b, 5c).

REVIEWERS' COMMENTS

Reviewer #1 (Remarks to the Author):

The authors have fully addressed my comments.

Reviewer #2 (Remarks to the Author):

The authors have thoroughly addressed all my comments and added an impressive amount of new data that strengthen the manuscript. Congratulations.

I only have a minor point: In Figure 2c, in the revised version they used the same PCGF1 ChIP-seq screen shots shown in the first version of the paper. This must be an error since they performed new PCGF1 ChIP-seq assays.

Reviewer #3 (Remarks to the Author):

I have read the revised manuscript version and the point to point response of the authors to criticisms. While they have in part addressed different specific points that were raised, it still remains unclear whether the transcriptional effects are directly inferred by PCGF1 loss or a consequence of differentiation failure. I do understand that this is a difficult question to address and limited to correlative data. Question 3 was indeed pointing at such specificity effects trying to determine why *Pcgf6* loss affected the expression of the same set of genes. The authors now show that loss of PCGF6 does not affect Ring1b binding at these genes during differentiation. This raises further concerns about the directness of these transcriptional effects since the same set of genes is similarly affected in transcription without any modulation in PcG activity (Fig2A vs. exFig2D-E). Specificity controls for PCGF1 ChIP analysis are also missing. Overall, the work provides a huge amount of information during early differentiation of ESC albeit finding and conclusions still provide very limited novelty remaining incremental respect to current knowledge.

Reviewer #1 (Remarks to the Author):

The authors have fully addressed my comments.

Reply) Thank you for this comment.

Reviewer #2 (Remarks to the Author):

The authors have thoroughly addressed all my comments and added an impressive amount of new data that strengthen the manuscript. Congratulations.

I only have a minor point: In Figure 2c, in the revised version they used the same PCGF1 ChIP-seq screen shots shown in the first version of the paper. This must be an error since their performed new PCGF1 ChIP-seq assays.

Reply) Thank you for your careful reading and a valuable comment. We have replaced the Figure 2c as you see in the attached appendix Figure 1. The same change is given to the revised manuscript.

Reviewer #3 (Remarks to the Author):

I have read the revised manuscript version and the point to point response of the authors to criticisms. While they have in part addressed different specific points that were raised, it still remains unclear whether the transcriptional effects are directly inferred by PCGF1 loss or a consequence of differentiation failure. I do understand that this is a difficult question to address and limited to correlative data. Question 3 was indeed

pointing at such specificity effects trying to determine why Pcgf6 loss affected the expression of the same set of genes. The authors now show that loss of PCGF6 does not affect Ring1b binding at these genes during differentiation. This raises further concerns about the directness of these transcriptional effects since the same set of genes is similarly affected in transcription without any modulation in PcG activity (Fig2A vs. exFig2D-E). Specificity controls for PCGF1 ChIP analysis are also missing. Overall, the work provides a huge amount of information during early differentiation of ESC albeit finding and conclusions still provide very limited novelty remaining incremental respect to current knowledge.

Reply) Thank you for your careful reading and valuable comments to our manuscript. We, particularly, appreciate the comment about the specificity of anti-PCGF1 antibody. I agree this is an important issue, however, the specificity of this antibody in ChIP analysis has been reported in our previous papers (Blackledge et al., 2014, Cell; Fursova et al., 2019, Molecular Cell). I have extracted related figures from these papers in the Appendix figure 2. Legends for this appendix is also attached.

(Ref)

Blackledge, N. P. *et al.* PRC1 Catalytic Activity Is Central to Polycomb System Function. *Mol. Cell* **77**, 857-874.e9 (2020).

Fursova, N. A. *et al.* Synergy between Variant PRC1 Complexes Defines Polycomb-Mediated Gene Repression. *Mol. Cell* **74**, 1020-1036.e8 (2019).

Briefly, in 2019 Fursova's Molecular Cell paper, we did ChIP-seq in the PCGF1/3/5 cKO cell line, in which we can delete PCGF1 together with PCGF3/5 by Tamoxifen-dependent activation of CRE recombinase (Appendix Figure 2A and 2B). Both the untreated and tamoxifen-treated data is in the GEO accession (GSE119618). I have made snapshots for two very strong Polycomb targets (HoxD locus and Lhx9) which normally have very high levels of PCGF1 (Appendix Figure 2B). As you can see that data looks very clean, with strong PCGF1-specific enrichment completely disappearing following tamoxifen treatment. However, in this experimental setup, as PCGF3/5 are deleted together with PCGF1, we cannot formally exclude the possibility that our antibody cross-react with PCGF3/5. However, PCGF3/5 are accepted not to accumulate at CpG islands (Fursova et al., 2019). We, therefore, suggest that ChIP signals given by our anti-PCGF1 represent accumulation of PCGF1. To further validate this point, I've also included some data from our 2014 Cell paper in which we depleted PCGF1 via shRNA in a TetR-KDM2B cell line (Appendix Figure 2C, 2D and 2E). The ChIP-qPCR

data shows very nicely that in the control cell line, TetR-KDM2B recruits PCGF1 (and RING1B) to the TetO array. However, following PCGF1 depletion there is a loss of PCGF1 ChIP-qPCR signal (and a failure to recruit RING1B). Taken these together, we concluded the specificity of the antibody was sufficiently validated.

In the present study, as the specificity of this antibody is sufficiently validated, we did not perform additional ChIP-seq analysis by using PCGF1-KO ES cells, which should provide the best control. I, however, think the quality of our data in this present study could be assessed by comparing them with those of Fursova paper at the genes of interests (in this case, Klf4, Tbx3, Pdgfa, Sox4, Grhl2 and Gapdh)(Appendix Figure 2F GSE119618). In Fursova paper, PCGF1 signals, which were completely depleted upon tamoxifen-induced deletion of PCGF1/3/5, associated with CpG islands associated with these genes (Appendix Figure 2F GSE119618). In our data shown in Appendix Figure 2F, PCGF1 signals accumulated at CpG islands in ES cells and EB and overlapped with those identified by Fursova et al. (Appendix Figure 2F This study). I, therefore, think specificity of the antibody in our ChIP-seq experiments is sufficiently qualified.

Accordingly, we have added a sentence to state the origin of this antibody in page 6, lines 13-14 in the revised manuscript.

Appendix Figure 1

Figure 2C

Updated Figure 2C

Appendix Figure 2

(A)

(B)

(C)

(E)

(D)

(F)

Appendix Figure2: Validation of PCGF1-specific antibody

(A) Western blot analysis of PCGF1 and other Polycomb factors in a *Pcgf1/3/5* conditional knockout ESC line in which addition of tamoxifen (OHT) causes deletion of *Pcgf1*, 3 and 5. Data shows that following PCGF1 removal, there is a complete loss of western blot signal using the PCGF1-specific antibody.

(B) Genomic snapshots of two Polycomb target loci (*HoxD* cluster and *Lhx9*), showing ChIP-seq for PCGF1 in the *Pcgf1/3/5* conditional knockout line with and without tamoxifen treatment. Data shows that following deletion of *Pcgf1*, ChIP-seq enrichment using the PCGF1-specific antibody is lost. ChIP-seq for RING1B in wild type ESCs is also shown (all data from A and B taken from PMID: 31029541, GEO accession: GSE119618).

(C) Western blot analysis of PCGF1, RING1B and LAMIN in a PCGF1 knockdown cell line (PCGF1shRNA) and a control cell line (CTLshRNA). Data shows that following PCGF1 knockdown, there is a loss of western blot signal using the PCGF1-specific antibody.

(D) Schematic showing recruitment of the PCGF1-vPRC1 complex to a TetO array via a TetR-KDM2B fusion protein.

(E) ChIP-qPCR data for TetR, PCGF1 and RING1B in CTLshRNA;TetR-KDM2B and PCGF1shRNA;TetR-KDM2B cell lines. Data shows that in the CTLshRNA cell line, TetR-KDM2B recruits both PCGF1 and RING1B to the TetO array. However, in the PCGF1shRNA cell line, knockdown of PCGF1 results in loss of PCGF1 ChIP-qPCR signal from the TetO array and a failure to recruit RING1B (data from C and E taken from PMID: 24856970).

(F) ChIP-Seq data for PCGF1 in *Pcgf1/3/5* triple conditional knockout ESC (Fursova et al., 2019, GSE119618) and this study for PCGF1 in ESC and EB. The sequence reads were aligned to mm10 and analyzed following Fursova et al., 2019.